# THE ROLE OF IMAGENET CLASSES IN FRÉCHET INCEPTION DISTANCE

**Tuomas Kynkäänniemi**
Aalto University
`tuomas.kynkaanniemi@aalto.fi`

**Tero Karras**
NVIDIA
`tkarras@nvidia.com`

**Miika Aittala**
NVIDIA
`maittala@nvidia.com`

**Timo Aila**
NVIDIA
`taila@nvidia.com`

**Jaakko Lehtinen**
Aalto University & NVIDIA
`jlehtinen@nvidia.com`

## ABSTRACT

Fréchet Inception Distance (FID) is the primary metric for ranking models in data-driven generative modeling. While remarkably successful, the metric is known to sometimes disagree with human judgement. We investigate a root cause of these discrepancies, and visualize what FID "looks at" in generated images. We show that the feature space that FID is (typically) computed in is so close to the ImageNet classifications that aligning the histograms of Top-$N$ classifications between sets of generated and real images can reduce FID substantially — without actually improving the quality of results. Thus, we conclude that FID is prone to intentional or accidental distortions. As a practical example of an accidental distortion, we discuss a case where an ImageNet pre-trained FastGAN achieves a FID comparable to StyleGAN2, while being worse in terms of human evaluation.

## 1 INTRODUCTION

Generative modeling has been an extremely active research topic in recent years. Many prominent model types, such as generative adversarial networks (GAN) (Goodfellow et al., 2014), variational autoencoders (VAE) (Kingma & Welling, 2014), autoregressive models (van den Oord et al., 2016b;a), flow models (Dinh et al., 2017; Kingma & Dhariwal, 2018) and diffusion models (Sohl-Dickstein et al., 2015; Song & Ermon, 2019; Ho et al., 2020) have seen significant improvement. Additionally, these models have been applied to a rich set of downstream tasks, such as realistic image synthesis (Brock et al., 2019; Razavi et al., 2019; Esser et al., 2021; Karras et al., 2019; 2020b;a; 2021), unsupervised domain translation (Zhu et al., 2017; Choi et al., 2020; Kim et al., 2020), image super resolution (Ledig et al., 2017; Bell-Kligler et al., 2019; Saharia et al., 2021), image editing (Park et al., 2019; 2020; Huang et al., 2022) and generating images based on a text prompt (Ramesh et al., 2021; Nichol et al., 2022; Ramesh et al., 2022; Saharia et al., 2022).

Given the large number of applications and rapid development of the models, designing evaluation metrics for benchmarking their performance is an increasingly important topic. It is crucial to reliably rank models and pinpoint improvements caused by specific changes in the models or training setups. Ideally, a generative model should produce samples that are indistinguishable from the training set, while covering all of its variation. To quantitatively measure these aspects, numerous metrics have been proposed, including Inception Score (IS) (Salimans et al., 2016), Fréchet Inception Distance (FID) (Heusel et al., 2017), Kernel Inception Distance (KID) (Binkowski et al., 2018), and Precision/Recall (Sajjadi et al., 2018; Kynkäänniemi et al., 2019; Naeem et al., 2020). Among these metrics, FID continues to be the primary tool for quantifying progress.

The key idea in FID (Heusel et al., 2017) is to separately embed real and generated images to a *vision-relevant feature space*, and compute a distance between the two distributions, as illustrated in Figure 1. In practice, the feature space is the penultimate layer (*pool3*, 2048 features) of an ImageNet (Deng et al., 2009) pre-trained Inception-V3 classifier network (Szegedy et al., 2016), and the distance is computed as follows. The distributions of real and generated embeddings are separately approximated by multivariate Gaussians, and their alignment is quantified using the Fréchet

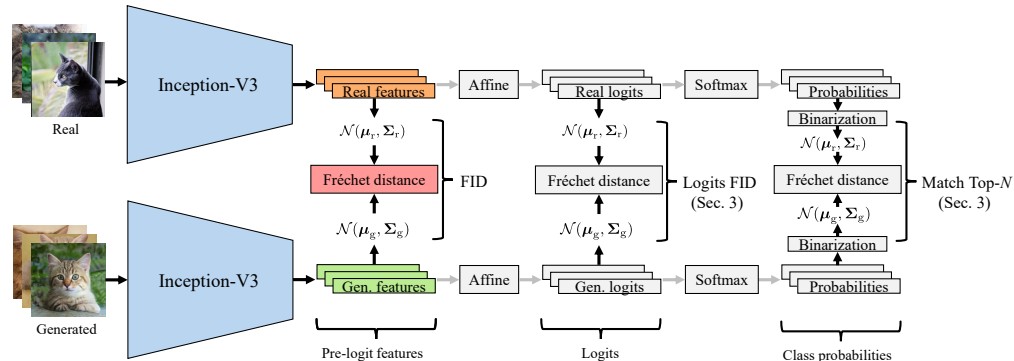

Figure 1: Overview of the Fréchet Inception Distance (FID) (Heusel et al., 2017). First, the real and generated images are separately passed through a pre-trained classifier network, typically the Inception-V3 (Szegedy et al., 2016), to produce two sets of feature vectors. Then, both distributions of features are approximated with multivariate Gaussians, and FID is defined as the Fréchet distance between the two Gaussians. In Section 3, we will compute alternative FIDs in the feature spaces of logits and class probabilities, instead of the usual pre-logit space.

(equivalently, the 2-Wasserstein or earth mover's) distance (Dowson & Landau, 1982)

$$\text{FID}\left(\boldsymbol{\mu}_{\text{r}}, \boldsymbol{\Sigma}_{\text{r}}, \boldsymbol{\mu}_{\text{g}}, \boldsymbol{\Sigma}_{\text{g}}\right) = \|\boldsymbol{\mu}_{\text{r}} - \boldsymbol{\mu}_{\text{g}}\|_2^2 + \text{Tr}\left(\boldsymbol{\Sigma}_{\text{r}} + \boldsymbol{\Sigma}_{\text{g}} - 2\left(\boldsymbol{\Sigma}_{\text{r}}\boldsymbol{\Sigma}_{\text{g}}\right)^{\frac{1}{2}}\right), \tag{1}$$

where $(\boldsymbol{\mu}_{\text{r}}, \boldsymbol{\Sigma}_{\text{r}})$, and $(\boldsymbol{\mu}_{\text{g}}, \boldsymbol{\Sigma}_{\text{g}})$ denote the sample mean and covariance of the embeddings of the real and generated data, respectively, and $\text{Tr}(\cdot)$ indicates the matrix trace. By measuring the distance between the real and generated embeddings, FID is a clear improvement over IS that ignores the real data altogether. FID has been found to correlate reasonably well with human judgments of the fidelity of generated images (Heusel et al., 2017; Xu et al., 2018; Lucic et al., 2018), while being conceptually simple and fast to compute.

Unfortunately, FID conflates the resemblance to real data and the amount of variation to a single value (Sajjadi et al., 2018; Kynkäänniemi et al., 2019), and its numerical value is significantly affected by various details, including the sample count (Binkowski et al., 2018; Chong & Forsyth, 2020), the exact instance of the feature network, and even low-level image processing (Parmar et al., 2022). Appendix A gives numerical examples of these effects. Furthermore, several authors (Karras et al., 2020b; Morozov et al., 2021; Nash et al., 2021; Borji, 2022; Alfarra et al., 2022) observe that there exists a discrepancy in the model ranking between human judgement and FID in non-ImageNet data, and proceed to introduce alternative metrics. Complementary to these works, we focus on elucidating *why* these discrepancies exist and what exactly is the role of ImageNet classes.

The implicit assumption in FID is that the feature space embeddings have general perceptual relevance. If this were the case, an improvement in FID would indicate a corresponding perceptual improvement in the generated images. While feature spaces with approximately this property have been identified (Zhang et al., 2018), there are several reasons why we doubt that FID's feature space behaves like this. *First*, the known perceptual feature spaces have very high dimensionality ($\sim$6M), partially because they consider the spatial position of features in addition to their presence. Unfortunately, there may be a contradiction between perceptual relevance and distribution statistics. It is not clear how much perceptual relevance small feature spaces (2048D for FID) can have, but it is also hard to see how distribution statistics could be compared in high-dimensional feature spaces using a finite amount of data. *Second*, FID's feature space is specialized to ImageNet classification, and it is thus allowed to be blind to any image features that fail to help with this goal. *Third*, FID's feature space ("pre-logits") is only one affine transformation away from the logits, from which a softmax produces the ImageNet class probabilities. We can thus argue that the features correspond almost directly to ImageNet classes (see Appendix B). *Fourth*, ImageNet classifiers are known to base their decisions primarily on textures instead of shapes (Geirhos et al., 2019; Hermann et al., 2020).

Together, these properties have important practical consequences that we set out to investigate. In Section 2 we use a gradient-based visualization technique, Grad-CAM (Selvaraju et al., 2017), to

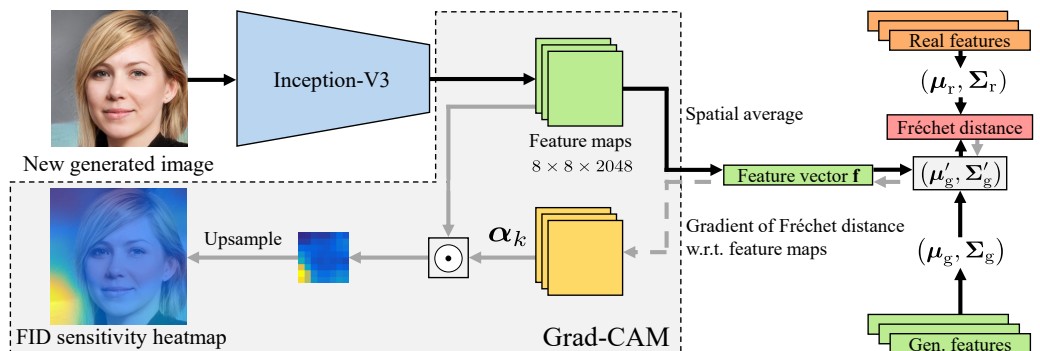

Figure 2: Visualizing which regions of an image FID is the most sensitive to. We augment the pre-computed feature statistics with a newly generated image, compute the FID, and use Grad-CAM (Selvaraju et al., 2017) to visualize the spatial importance in low-resolution feature maps that are subsequently upsampled to match the input resolution.

visualize what FID "looks at" in generated images, and observe that its fixation on the most prominent ImageNet classes makes it more interested in, for example, seat belts and suits, than the human faces in FFHQ (Karras et al., 2019). It becomes clear that when a significant domain gap exists between a dataset of interest and ImageNet, many of the activations related to ImageNet class templates are rather coincidental. We call such poorly fitting templates *fringe features* or fringe classes. As matching the distribution of such fringe classes between real and generated images becomes an obvious way of manipulating FID, we examine such "attacks" in detail in Section 3. The unfortunate outcome is that FID can be significantly improved – with hardly any improvement in the generated images – by selecting a subset of images that happen to match some number of fringe features with the real data. We conclude with an example of practical relevance in Section 4, showing that FID can be unreliable when ImageNet pre-trained discriminators (Sauer et al., 2021; Kumari et al., 2022) are used in GANs. Some of the improvement in FID comes from accidental leaking of ImageNet features, and the consequent better reproduction of ImageNet-like aspects in the real data. We hope that the new tools we provide open new opportunities to better understand the existing evaluation metrics and develop new ones in the future. Code is available at https://github.com/kynkaat/role-of-imagenet-classes-in-fid.

## 2 What does FID look at in an image?

We will now inspect which parts of an image FID is the most sensitive to. We rely on Grad-CAM (Selvaraju et al., 2017) that has been extensively used for visualizing the parts of an image that contribute the most to the decisions of a classifier. We want to use it similarly for visualizing which parts of one generated image contribute the most to FID. A key challenge is that FID is defined only between large sets of images (50k), not for a single image. We address this difficulty by pre-computing the required statistics for a set of 49,999 generated images, and augmenting them with the additional image of interest. We then use Grad-CAM to visualize the parts of the image that have the largest influence on FID. We will first explain our visualization technique in more detail, followed by observations from individual images, and from aggregates of images.

**Our visualization technique** Figure 2 gives an outline of our visualization technique. Assume we have pre-computed the mean and covariance statistics $(\boldsymbol{\mu}_\text{r}, \boldsymbol{\Sigma}_\text{r})$ and $(\boldsymbol{\mu}_\text{g}, \boldsymbol{\Sigma}_\text{g})$ for 50,000 real and 49,999 generated images, respectively. These Gaussians are treated as constants in our visualization. Now, we want to update the statistics of the generated images by adding one new image. Computing the FID from the updated statistics allows us to visualize which parts of the added image influence it the most. Given an image, we feed it through the Inception-V3 network to get activations $\boldsymbol{A}^k$ before the *pool3* layer. The spatial resolution here is $8 \times 8$ and there are 2048 feature maps. The spatial averages of these feature maps correspond to the 2048-dimensional feature space where FID is calculated. We then update the pre-computed statistics $(\boldsymbol{\mu}_\text{g}, \boldsymbol{\Sigma}_\text{g})$ by including the features $\boldsymbol{f}$ of the new sample (Pebay, 2008): $(\boldsymbol{\mu}_\text{g}' = \frac{N-1}{N}\boldsymbol{\mu}_\text{g} + \frac{1}{N}\boldsymbol{f}, \boldsymbol{\Sigma}_\text{g}' = \frac{N-2}{N-1}\boldsymbol{\Sigma}_\text{g} + \frac{1}{N}(\boldsymbol{f} - \boldsymbol{\mu}_\text{g})^T(\boldsymbol{f} - \boldsymbol{\mu}_\text{g}))$. Here, $N = 50,000$ is the size of the updated set. To complete the forward pass, we evaluate FID using these modified statistics.

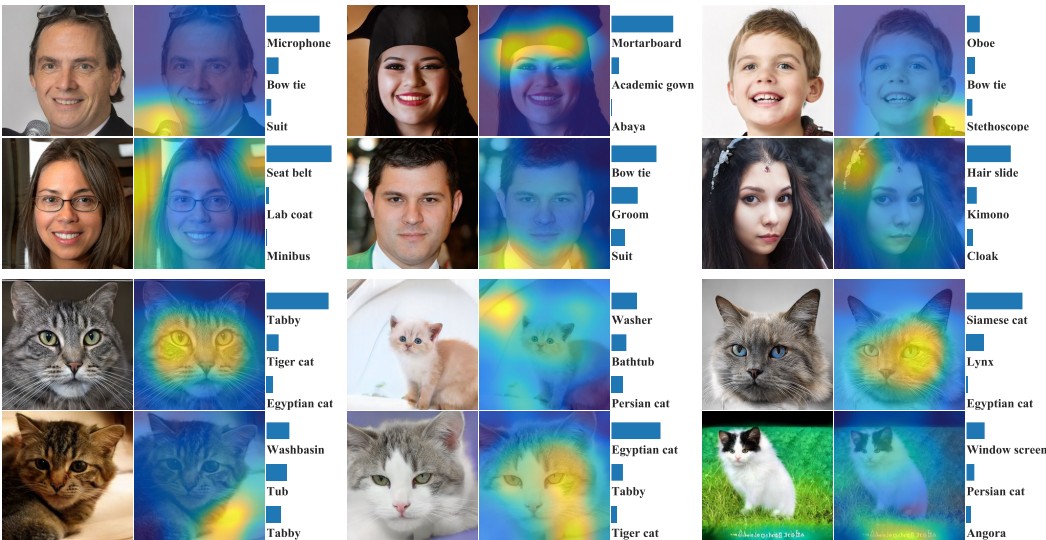

Figure 3: StyleGAN2 generated images along with heatmap visualizations of the image regions that FID considers important in FFHQ (top) and LSUN CAT (bottom). Yellow indicates regions that are more important and blue regions that are less important, i.e., modifying the content of the yellow regions affects FID most strongly. As many of the yellow areas are completely outside the intended subject, we sought an explanation from the ImageNet Top-3 class predictions. FID is very strongly focused on the area that corresponds to the predicted Top-1 class — whatever that may be. We discuss the qualitative difference between FFHQ and LSUN CAT in the main text.

In a backward pass, we first estimate the importance $\boldsymbol{\alpha}_k$ for each of the $k$ feature maps as

$$\boldsymbol{\alpha}_k = \frac{1}{8 \times 8} \sum_i \sum_j \left| \frac{\partial \mathrm{FID}(\boldsymbol{\mu}_\mathrm{r}, \boldsymbol{\Sigma}_\mathrm{r}, \boldsymbol{\mu}'_\mathrm{g}, \boldsymbol{\Sigma}'_\mathrm{g})}{\partial \boldsymbol{A}^k_{ij}} \right|^2 \tag{2}$$

Then, an $8 \times 8$-pixel spatial importance map is computed as a linear combination $\sum_k \boldsymbol{\alpha}_k \boldsymbol{A}^k$. Note that Selvaraju et al. (2017) originally suggested using a ReLU to only visualize the regions that have a positive effect on the probability of a given class; we do not employ this trick, since we are interested in visualizing both positive and negative effects on FID. Additionally, we upsample the importance map using Lanczos filtering to match the dimensions of the input image. Finally, we convert the values of the importance map to a heatmap visualization. See Appendix C for comparisons of our FID heatmaps and standard Grad-CAM.

**Observations from individual images**   Figure 3 shows heatmaps of the most important regions for FID in FFHQ (Karras et al., 2019) and LSUN CAT (Yu et al., 2015). Given that in FFHQ the goal is to generate realistic human faces, the regions FID considers most important seem rather unproductive. They are typically outside the person's face. To better understand why this might happen, recall that ImageNet does not include a "person" or a "face" category. Instead, some other fringe features (and thus classes) are necessarily activated for each generated image. Interestingly, we observe that the heatmaps seem to correspond very well with the areas that the image's ImageNet Top-1 class prediction would occupy. Note that these ImageNet predictions were not used when computing the heatmap; they are simply a tool for understanding what FID was focusing on. As a low FID mandates that the distributions of the most prominent fringe classes are matched between real and generated images, one has to wonder how relevant it really is to match the distributions of "seat belt" or "oboe" in this dataset. In any case, managing to perform such matching would seem to open a possible loophole in FID; we will investigate this further in Section 3.

In contrast to human faces, ImageNet does include multiple categories of cats. This helps FID to much better focus on the subject matter in LSUN CAT, although some fringe features from the image background are still getting picked up, and a very low FID would require that "washbasin", etc. are similarly detected in real and generated images.

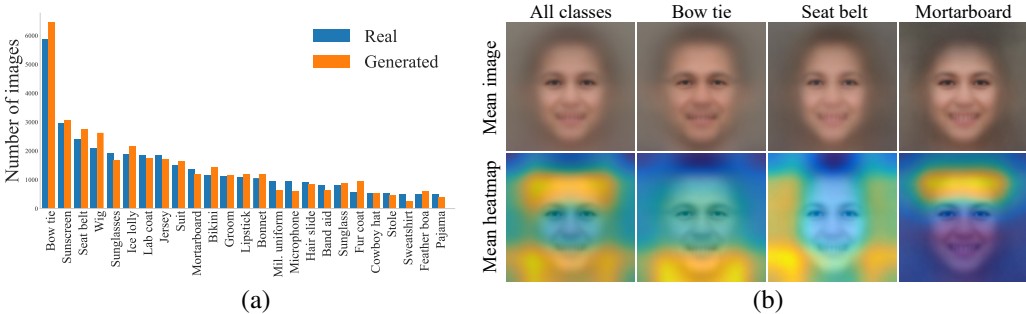

Figure 4: (a) Distribution of the ImageNet Top-1 classes, predicted by Inception-V3, for real and StyleGAN2 generated images in FFHQ. (b) Mean images and Grad-CAM heatmaps among all classes, and images classified as "bow tie", "seat belt" and "mortarboard". These were computed from generated images. Strikingly, when averaging over all classes, FID is the most sensitive to ImageNet objects that are located outside the face area.

**Observations from aggregates of images**   Figure 4a shows the distribution of ImageNet Top-1 classifications for real and generated images in FFHQ. Typically, the predicted classes are accessories, but their presence might be correlated with persons in the images. In a further test, we calculated a heatmap of the average sensitivity over 50k generated images. Figure 4b shows average images and heatmaps for all classes, and for images that are classified to some specific class (e.g., "bow tie"). If we average over all classes, FID is the most sensitive to other areas of the image than the human face. If we compute a heatmap of the average sensitivity within a class, they highlight the regions where the corresponding ImageNet class is typically located in the images. Appendix C provides additional single image and average heatmaps. It also provides further validation that the image regions FID is the most sensitive to are correctly highlighted by our approach.

In related work, van Steenkiste et al. (2020) found that in images with multiple salient objects, FID focuses on only one of them and hypothesized that the metric could be easily fooled by a generative model that focuses on some image statistics (e.g., generating the correct number of objects) rather than content. They believe that this is because Inception-V3 is trained on a single object classification.

## 3   PROBING THE PERCEPTUAL NULL SPACE IN FID

Our findings with Grad-CAM raise an interesting question: to what extent could FID be improved by merely nudging the generator to produce images that get classified to the same ImageNet classes as the real data? As we are interested in a potential weakness in FID, we furthermore want this nudging to happen so that the generated results do *not* actually improve in any real sense. In our terminology, operations that change FID without changing the generated results in a perceptible way are exploiting the *perceptual null space* of FID.

We will first test a simple Top-1 (ImageNet classification) histogram matching between real and generated data. As this has only a modest impact on FID, we proceed to develop a more general distribution resampling method. This approach manages to reduce FID very significantly, indicating that there exists a large perceptual null space in FID. We then explore the role of other likely class labels, in addition to the Top-1 label, by aligning the Top-$N$ histograms. In Section 4 we observe that this theoretical weakness also has practical relevance.

In our experiments, we use StyleGAN2 auto-config trained in $256 \times 256$ resolution without adaptive discriminator augmentation (ADA). The only exception is AFHQ-v2 DOG, where we enable ADA and train in $512 \times 512$ resolution.[1] Following standard practice, we compute FID against the training set, using 50k randomly chosen real and generated images and the official TensorFlow version of Inception-V3.[2] A difference between our FIDs for StyleGAN2 and the ones reported by Karras et al. (2020a) is caused by the use of different training configurations.

---

[1] We use the official code `https://github.com/NVlabs/stylegan2-ada-pytorch`.

[2] `http://download.tensorflow.org/models/image/imagenet/inception-2015-12-05.tgz`

Table 1: Results of Top-1 histogram matching. We compare the FID of randomly sampled images (FID) against ones that have been resampled to match the Top-1 histogram of the training data (FID$^{\text{Top-1}}$). The numbers represent averages over five FID evaluations. Additionally, we report the corresponding numbers by replacing the Inception-V3 feature space with ResNet-50 (FID$_{\text{ResNet-50}}$), SwAV (FID$_{\text{SwAV}}$), and CLIP features (FID$_{\text{CLIP}}$). Note that we use these alternative feature spaces only when computing FID; the resampling is still done using Inception-V3. The numerical values between different features spaces are not comparable.

| Dataset | FID | FID$^{\text{Top-1}}$ | FID$_{\text{ResNet-50}}$ | FID$_{\text{ResNet-50}}^{\text{Top-1}}$ | FID$_{\text{SwAV}}$ | FID$_{\text{SwAV}}^{\text{Top-1}}$ | FID$_{\text{CLIP}}$ | FID$_{\text{CLIP}}^{\text{Top-1}}$ |
|---|---|---|---|---|---|---|---|---|
| FFHQ | 5.30 | 4.70 $(-11.3\%)$ | 6.11 | 5.59 $(-8.5\%)$ | 1.42 | 1.41 $(-0.7\%)$ | 2.76 | 2.74 $(-0.7\%)$ |
| LSUN Cat | 8.25 | 7.37 $(-10.7\%)$ | 12.33 | 11.29 $(-8.4\%)$ | 2.99 | 2.96 $(-1.0\%)$ | 8.94 | 8.83 $(-1.2\%)$ |
| LSUN Car | 5.65 | 5.17 $(-8.5\%)$ | 8.79 | 8.51 $(-3.2\%)$ | 2.39 | 2.38 $(-0.4\%)$ | 7.75 | 7.73 $(-0.3\%)$ |
| LSUN Places | 12.96 | 11.76 $(-9.3\%)$ | 15.20 | 13.61 $(-10.5\%)$ | 3.12 | 3.02 $(-3.2\%)$ | 16.35 | 16.17 $(-1.1\%)$ |
| AFHQ-v2 Dog | 10.25 | 9.39 $(-8.4\%)$ | 13.71 | 13.19 $(-3.8\%)$ | 2.78 | 2.77 $(-0.4\%)$ | 4.25 | 4.16 $(-2.1\%)$ |

**Top-1 histogram matching** Based on the Grad-CAM visualizations in Section 2, one might suspect that to achieve a low FID, it would be sufficient to match the Top-1 class histograms between the sets of real and generated images. We tested this hypothesis by computing the Top-1 histogram for 50k real images, and then sampling an equal number of unique generated images for each Top-1 class. This was done by looking at the class probabilities at the output of the Inception-V3 classifier (Figure 1), and discarding the generated images that fall into a bin that is already full. Over multiple datasets, this simple Top-1 histogram matching consistently improves FID, by $\sim 10\%$ (Table 1).

Does this mean that the set of generated images actually improved? A genuine improvement should also be clearly visible in FIDs computed using alternative feature spaces. To this end, we calculated FIDs in the feature spaces of a ResNet-50 ImageNet classifier (FID$_{\text{ResNet-50}}$) (He et al., 2016), self-supervised SwAV classifier (FID$_{\text{SwAV}}$) (Caron et al., 2020; Morozov et al., 2021), and CLIP image encoder (FID$_{\text{CLIP}}$) (Radford et al., 2021; Sauer et al., 2021).[3] Interestingly, FID$_{\text{ResNet-50}}$ drops almost as much as the original FID, even though the resampling was carried out with the Inception-V3 features. This makes sense because both feature spaces are necessarily very sensitive to the ImageNet classes. FID$_{\text{SwAV}}$ drops substantially less, probably because it was never trained to classify the ImageNet data. The muted decrease in FID$_{\text{CLIP}}$ is also in line with expectations because CLIP never saw ImageNet data; it was trained with a different task of matching images with captions. [4] We can thus conclude that the observed decrease in FID is closely related to the degree of ImageNet pre-training, and that the alternative feature spaces fail to confirm a clear increase in the result quality.

As a ten percent reduction in FID may not be significant enough for a human observer to draw reliable conclusions from the sets of generated images, we proceed to generalize the histogram matching to induce a much larger drop in FID.

**Matching all fringe features** We will now design a general technique for resampling the distribution of generated images, with the goal of approximately matching *all* fringe features. We will subsequently modify this approach to do Top-$N$ histogram matching, as extending the simple "draw samples until it falls into the right bin"-approach for Top-$N$ would be computationally infeasible.

Our idea is to first generate a larger set of candidate images ($5\times$ oversampling), and then carefully select a subset of these candidates so that FID decreases. We approach this by directly optimizing FID as follows. First, we select a *candidate set* of 250k generated images and compute their Inception-V3 features. We then assign a non-negative scalar weight $w_i$ to each generated image, and optimize the weights to minimize FID computed from *weighted* means and covariances of the generated images. After optimization, we use the weights as sampling probabilities and draw 50k random samples with replacement from the set of 250k candidate images. More precisely, we optimize

$$\min_{\boldsymbol{w}} \left( \left\| \boldsymbol{\mu}_{\text{r}} - \boldsymbol{\mu}_{\text{g}}(\boldsymbol{w}) \right\|_2^2 + \text{Tr} \left( \boldsymbol{\Sigma}_{\text{r}} + \boldsymbol{\Sigma}_{\text{g}}(\boldsymbol{w}) - 2 \left( \boldsymbol{\Sigma}_{\text{r}} \boldsymbol{\Sigma}_{\text{g}}(\boldsymbol{w}) \right)^{\frac{1}{2}} \right) \right), \tag{3}$$

where $\boldsymbol{\mu}_{\text{g}}(\boldsymbol{w}) = \frac{\sum_i w_i \boldsymbol{f}_i}{\sum_i w_i}$ and $\boldsymbol{\Sigma}_{\text{g}}(\boldsymbol{w}) = \frac{1}{\sum_i w_i} \sum_i w_i \left( \boldsymbol{f}_i - \boldsymbol{\mu}_{\text{g}}(\boldsymbol{w}) \right)^T \left( \boldsymbol{f}_i - \boldsymbol{\mu}_{\text{g}}(\boldsymbol{w}) \right)$ are the weighted mean and covariance of generated features $\boldsymbol{f}_i$, respectively. In practice, we optimize the

---

[3]We use the ViT-B/32 model available in `https://github.com/openai/CLIP`.

[4]One may argue that FID$_{\text{CLIP}}$ is less sensitive and that $\sim 1\%$ decrease is significant. However, in Section 4, we show a case where FID$_{\text{CLIP}}$ decreases by $\sim 40\%$ and observe a qualitative improvement in the results.

Table 2: Results of matching all fringe features. We compare the FID of randomly sampled images (FID) against ones that have been resampled to approximately match all fringe features with the training data ($FID^{PL}$). The numbers represent averages over ten FID evaluations. $FID_{ResNet-50}$, $FID_{SwAV}$, $FID_{CLIP}$ match the descriptions in Table 1. The gray column ($FID^L$) shows an additional experiment where the resampling is done using logits instead of the usual pre-logits.

| Dataset | FID | $FID^{PL}$ | $FID^L$ | $FID_{ResNet-50}$ | $FID^{PL}_{ResNet-50}$ | $FID_{SwAV}$ | $FID^{PL}_{SwAV}$ | $FID_{CLIP}$ | $FID^{PL}_{CLIP}$ |
|---|---|---|---|---|---|---|---|---|---|
| FFHQ | 5.30 | 1.78 (−66.4%) | 2.22 (−58.1%) | 6.11 | 3.85 (−37.0%) | 1.42 | 1.24 (−12.7%) | 2.76 | 2.64 (−4.3%) |
| LSUN CAT | 8.25 | 3.05 (−63.0%) | 3.88 (−53.0%) | 12.33 | 7.17 (−41.8%) | 2.99 | 2.71 (−9.4%) | 8.94 | 7.98 (−10.7%) |
| LSUN CAR | 5.65 | 2.11 (−62.7%) | 2.59 (−54.2%) | 8.79 | 5.88 (−33.1%) | 2.39 | 2.15 (−10.0%) | 7.75 | 7.33 (−5.4%) |
| LSUN PLACES | 12.96 | 3.59 (−72.3%) | 4.43 (−65.8%) | 15.20 | 9.35 (−38.5%) | 3.12 | 2.60 (−16.7%) | 16.35 | 14.54 (−11.1%) |
| AFHQ-v2 DOG | 10.25 | 5.92 (−42.2%) | 6.27 (−38.8%) | 13.71 | 11.38 (−17.1%) | 2.78 | 2.62 (−5.8%) | 4.25 | 4.04 (−4.9%) |

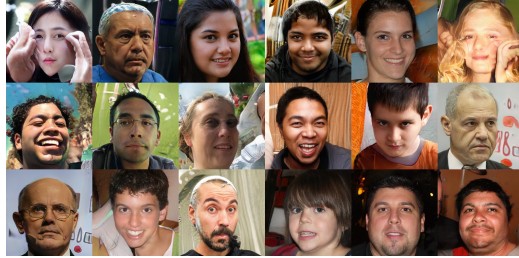 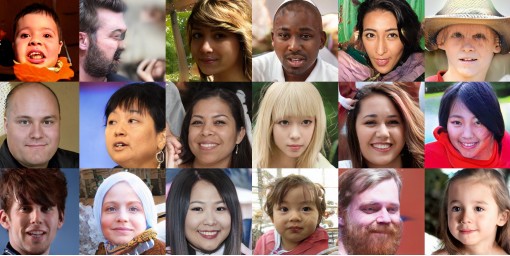

(a) Images with small weights (b) Images with large weights

Figure 5: Uncurated random StyleGAN2 samples from images with (a) the smallest 10% of weights and (b) the largest 10% of weights after optimizing the weights to improve FID. Both sets contain both realistic images and images with clear visual artifacts in roughly equal proportions. See Appendix D for a larger sample.

weights in Equation 3 via gradient descent. We parameterize the weights as $\log(w_i)$ to avoid negative values and facilitate easy conversion to probabilities. Note that we do not optimize the parameters of the generator network, only the sampling weights that are assigned to each generated image in the candidate set. Optimizing FID directly in this way is a form of adversarial attack, but not nearly as strong as modifying the images or the generator to directly attack the Inception-V3 network. In any case, our goal is only to elucidate the perceptual null space in FID, and we do not advocate this resampling step to improve the quality of models (Issenhuth et al., 2022; Humayun et al., 2022).

Table 2 shows that FIDs can be drastically reduced using this approach. An improvement by as much as 60% would be considered a major breakthrough in generative modeling, and it should be completely obvious when looking at the generated images. Yet, the uncurated grids in Appendix D fail to demonstrate an indisputable improvement. To confirm the visual result quantitatively, we again compute FIDs of the resampled distributions in the alternative feature spaces. We see a substantially smaller improvement in feature spaces that did not use ImageNet classifier pre-training. While it is possible that the generated results actually improved in some minor way, we can nevertheless conclude that a vast majority of the improvement in FID occurred in its perceptual null space, and that this null space is therefore quite large. In other words, FID can be manipulated to a great extent through the ImageNet classification probabilities, without meaningfully improving the generated results.

Figure 5 further shows images that obtain small and large weights in the optimization; it is not obvious that there is a visual difference between the sets, indicating that the huge improvement in FID (-66.4%) cannot be simply attributed to discarding images with clear artifacts. Appendix D also shows that the resampling fools KID just as thoroughly as FID, even though we do not directly optimize KID.

Finally, it makes only a small difference whether the weight optimization is done in the typical pre-logit space (denoted $FID^{PL}$) or in the logit space ($FID^L$), confirming that these spaces encode approximately the same information. Figure 1 illustrates the difference between these spaces.

**Top-$N$ histogram matching** The drastic FID reduction observed in the previous section provides clues about the upper bound of the size of the perceptual null space. We will now further explore the nature of this null space by extending our resampling method to approximate Top-$N$ histogram

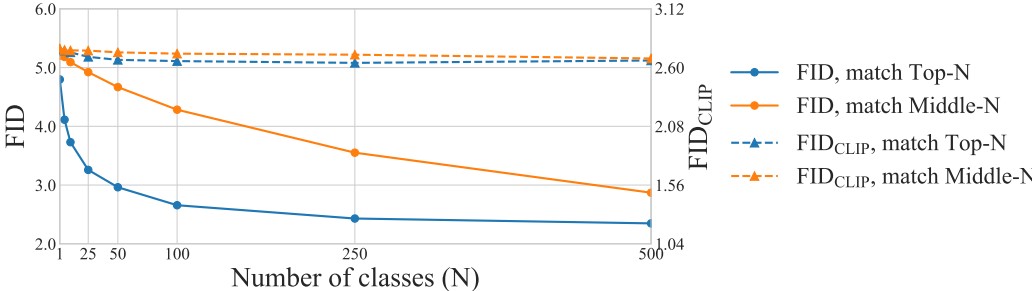

Figure 6: Softly matching Top-N class distributions in FFHQ through resampling. FID (solid curves, left-hand $y$ scale) decreases sharply with increasing number $N$ of classes included in the Top-N indicator vectors. At the same time, $\mathrm{FID}_{\mathrm{CLIP}}$ (dashed curves, right-hand $y$ scale) remains almost constant, indicating that the apparent improvements in FID are superfluous. The orange control curves have been computed from classes in the middle of the sorted probability vectors, indicating that the top classes indeed have a much stronger influence. As the numerical values of FID and $\mathrm{FID}_{\mathrm{CLIP}}$ are not comparable, the left and right $y$ axes have been normalized such that the relative changes are represented accurately.

matching. Then, by sweeping over $N$, we can gain further insights to how important the top classification results are for FID.

We implement this by carrying out the weight optimization in the space of class probabilities (see Figure 1). We furthermore binarize the class probability vectors by identifying the $N$ classes with the highest probabilities and setting the corresponding entries to 1 and the rest to 0. These vectors now indicate, for each image, the Top-$N$ classes, while discarding their estimated probabilities. By matching the statistics of these indicator vectors between real and generated distributions, we optimize the co-occurrence of Top-$N$ classes. The result of the weight optimization is therefore an approximation of Top-$N$ histogram matching. With $N = 1$, the results approximately align with simple histogram matching (Table 1). Note that the binarization is done before the weight optimization begins, and thus we don't need its (non-computable) gradients.

Figure 6 shows how FID (computed in the usual pre-logit space) changes as we optimize Top-$N$ histogram matching with increasing $N$. We observe that even with small values of $N$, FID improves rapidly, and converges to a value slightly higher than was obtained by optimizing the weights in the pre-logit space ($\mathrm{FID}^{\mathrm{PL}}$ in Table 2). This demonstrates that FID is, to a significant degree, determined by the co-occurrence of top ImageNet classes. Furthermore, it illustrates that FID is the most interested in a handful of features whose only purpose is to help with ImageNet classification, not on some careful analysis of the whole image. As a further validation of this tendency, we also computed a similar optimization using binarized vectors computed using $N$ classes chosen from the middle[5] of the sorted probabilities (orange curves in Figure 6). The results show that the top classes have a significantly higher influence on FID than those ranked lower by the Inception-V3 classifier. Finally, as before, we present a control $\mathrm{FID}_{\mathrm{CLIP}}$ (dashed curves) that shows that CLIP's feature space is almost indifferent to the apparent FID improvements yielded by the better alignment of Top-$N$ ImageNet classes. Though we only present results for FFHQ here, qualitative behavior is similar for LSUN CAT/PLACES/CAR, and AFHQ-V2 DOG (see Appendix D).

## 4 PRACTICAL EXAMPLE: IMAGENET PRE-TRAINED GANS

Our experiments indicate that it is certainly possible that some models receive unrealistically low FID simply because they happen to reproduce the (inconsequential) ImageNet class distribution detected in the training data. Perhaps the most obvious way this could happen in practice is when ImageNet pre-training is used for a GAN discriminator (Sauer et al., 2021; Kumari et al., 2022). This approach has been observed to lead to much faster convergence and significant improvements in FID, but since the discriminator is readily sensitive to the ImageNet classes, maybe it also guides the generator to replicate them? Perhaps a part of the improvement is in the perceptual nullspace of FID?

---

[5]Our binarization works when $\leq 50\%$ of classes are set to 1. After that the roles of 0 and 1 flip and the FID curves repeat themselves symmetrically. That is why we used middle entries instead of the lowest entries.

FID = 5.28, Recall = 0.45, FID$_{\text{CLIP}}$ = 4.67        FID = 5.30, Recall = 0.46, FID$_{\text{CLIP}}$ = 2.76

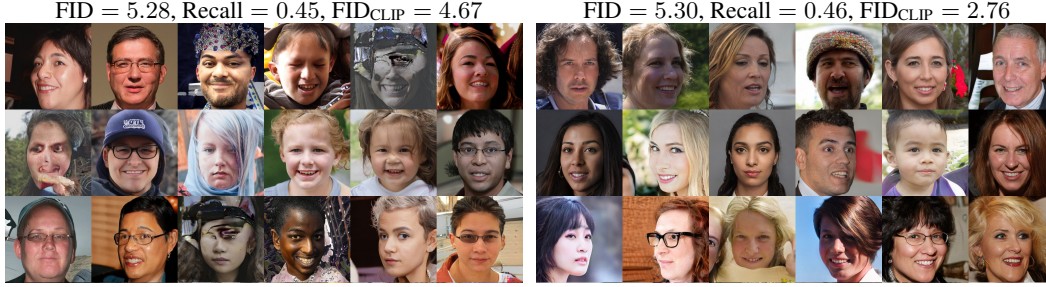

(a) Projected FastGAN                (b) StyleGAN2

Figure 7: Uncurated samples from (a) Projected FastGAN and (b) StyleGAN2. Both models achieve similar FID even though the Projected FastGAN samples contain more artifacts. In contrast, Projected FastGAN has significantly higher FID$_{\text{CLIP}}$, consistent with the observed quality differential.

We study this in the same context as Sauer et al. (2021) by training a Projected FastGAN (Liu et al., 2021; Sauer et al., 2021) that uses an ImageNet pre-trained EfficientNet (Tan & Le, 2019) as a feature extractor of the discriminator, and compare it against StyleGAN2 in FFHQ.[6]

The human preference study conducted by Sauer et al. (2021) concludes that despite the dramatic improvements in FID, Projected FastGAN tends to generate lower quality and less diverse FFHQ samples than StyleGAN2. We verify this by comparing samples from Projected FastGAN and Style-GAN2 in a setup where FIDs are roughly comparable and the models reproduce a similar degree of variation as measured by Recall (Kynkäänniemi et al., 2019). Visual inspection of uncurated samples (Figure 7) indeed reveals that Projected FastGAN produces much more distortions in the human faces than StyleGAN2 (see Appendix E for larger image grids).

In this iso-FID comparison, FID$_{\text{CLIP}}$ agrees with human assessment – StyleGAN2 is rated significantly better than Projected FastGAN. It therefore seems clear that Projected FastGAN has lower FID than it should have, confirming that at least some of its apparent improvements are in the perceptual null space. We believe that the reason for this is the accidental leak of information from the pre-trained network, causing the model to replicate the ImageNet-like aspects in the training data more keenly. This observation does not mean that ImageNet pre-training is a bad idea, but it does mean that such pre-training can make FID unreliable in practice.

We suspect similar interference can happen when the ImageNet pre-trained classifiers are used to curate the training data (DeVries et al., 2020) or as a part of the sampling process (Watson et al., 2022).

## 5   CONCLUSIONS

The numerical values of FID have a number of important uses. Large values indicate training failures quite reliably, and FID appears highly dependable when monitoring the convergence of a training run. FID improvements obtained through hyperparameter sweeps or other trivial changes generally seem to translate to better (subjective) results, even when the distributions are well aligned.[7] The caveats arise when two sufficiently different architectures and/or training setups are compared. If one of them is, for some reason, inclined to better reproduce the fringe features, it can lead to a much lower FIDs without a corresponding improvement in the human-observable quality.

Particular care should be exercised when introducing ImageNet pre-training to generative models (Sauer et al., 2021; 2022; Kumari et al., 2022), as it may compromise the validity of FID as a quality metric. This effect is difficult to quantify because the current widespread metrics (KID and Precision/Recall) also rely on the feature spaces of ImageNet classifiers. As a partial solution, the FID improvements should at least be verified using a non-ImageNet trained Fréchet distance. Viable alternative feature spaces include CLIP (Radford et al., 2021; Sauer et al., 2021), self-supervised SwAV (Caron et al., 2020; Morozov et al., 2021), and an uninitialized network (Naeem et al., 2020; Sauer et al., 2022). We hope that our methods help examine the properties of these feature spaces in future work.

---

[6]We use `https://github.com/autonomousvision/projected_gan` with default parameters.
[7]Based on personal communication with individuals who have trained over 10,000 generative models.

ACKNOWLEDGEMENTS

We thank Samuli Laine for helpful comments. This work was partially supported by the European Research Council (ERC Consolidator Grant 866435), and made use of computational resources provided by the Aalto Science-IT project and the Finnish IT Center for Science (CSC).

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

| Sample size | FFHQ | LSUN Cat |
|:---:|:---:|:---:|
| 5k + 5k | $11.65 \pm 0.04$ | $17.38 \pm 0.13$ |
| 10k + 10k | $8.31 \pm 0.06$ | $12.42 \pm 0.06$ |
| 50k + 50k | $5.30 \pm 0.04$ | $8.25 \pm 0.04$ |
| All + 50k | $5.14 \pm 0.04$ | $7.83 \pm 0.03$ |

|  | FFHQ | LSUN Cat |
|:---:|:---:|:---:|
| TensorFlow | $5.30 \pm 0.04$ | $8.25 \pm 0.04$ |
| PyTorch | $3.69 \pm 0.05$ | $6.64 \pm 0.03$ |

(a) Number of samples        (b) Inception-V3 instance

Figure 8: FID is very sensitive to (a) the number samples (real + generated) and (b) the exact instance of the Inception-V3 network. The tables report the mean $\pm$ standard deviation of FID for a given StyleGAN2 generator over ten evaluations with different random seeds.

## A NUMERICAL SENSITIVITY OF FID

Previous work has uncovered various details that have a surprisingly large effect on the exact value of FID (Binkowski et al., 2018; Lucic et al., 2018; Parmar et al., 2022; Chong & Forsyth, 2020). These observations are important to acknowledge because reproducing results from comparison methods is not always possible and one might be forced to resort to copying the reported FID results. In that case, even small differences in the FID evaluation protocol might cause erroneous rankings between models.

**Number of samples and bias.** FID depends strongly on the number of samples used in evaluation (Figure 8a). Therefore it is crucial to standardize to a specific number of real and generated samples (Binkowski et al., 2018).

**Network architecture.** FID is also very sensitive to the chosen feature network instance or type. Many deep learning frameworks (e.g. PyTorch (Paszke et al., 2019), Tensorflow (Abadi et al., 2015)) provide their own versions of the Inception-V3 network with distinct weights. Figure 8b shows that FID is surprisingly sensitive to the exact instance of the Inception-V3 network. The discrepancies are certainly large enough to confuse with state-of-the-art performance. In practice, the official Tensorflow network must be used for comparable results.[8]

Lucic et al. (2018) reported that the *ranking* of models with FID is not sensitive to the selected network architecture – it only has an effect on the absolute value range of FIDs, but the relative ordering of the models remains approximately the same. However, our tests with $FID_{CLIP}$ indicate that this observation likely holds only between different ImageNet classifier networks.

**Image processing flaws.** Before feeding images to the Inception-V3, they need to be resized to $299 \times 299$ resolution. Parmar et al. (2022) noted that the image resize functions of the most commonly used deep learning libraries (Paszke et al., 2019; Abadi et al., 2015) introduce aliasing artifacts due to poor pre-filtering. They demonstrate that this aliasing has a noticeable effect on FID.

---

[8] http://download.tensorflow.org/models/image/imagenet/inception-2015-12-05.tgz

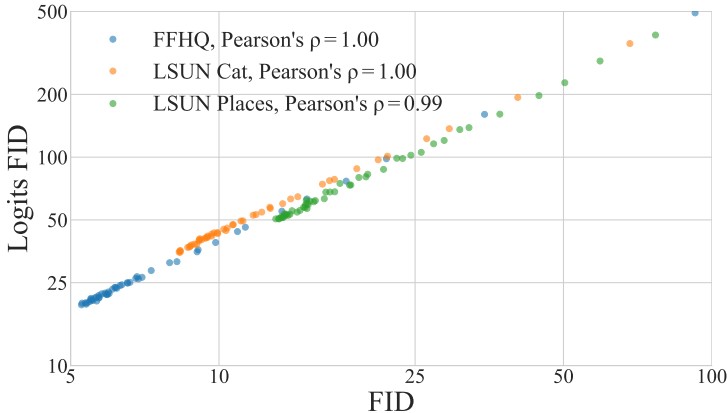

Figure 9: We observe nearly perfect correlation between FIDs computed from pre-logit features and classification logits since these two are separated by only one affine transformation. Each point corresponds to a single StyleGAN2 training snapshot in $256 \times 256$ resolution.

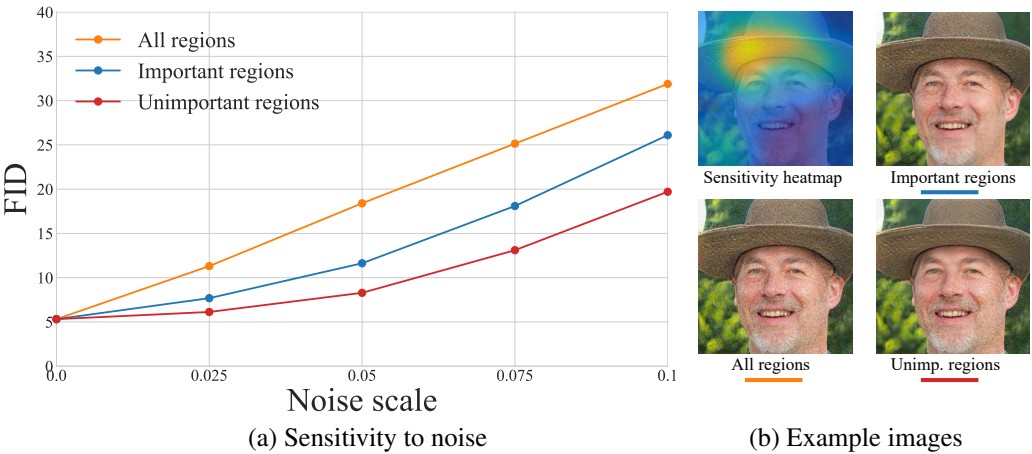

(a) Sensitivity to noise                                    (b) Example images

Figure 10: (a) Adding Gaussian noise to regions that are important for FID (blue curve) leads to the larger increase compared to adding noise in unimportant regions. (b) Image showing regions FID considers as the most important and noise added to different regions at scale 0.05. We recommend zooming in (b) to better assess the noise.

## B    CORRELATION BETWEEN PRE-LOGITS AND LOGITS FID

Figure 9 demonstrates that FIDs calculated from the pre-logits and logits are highly correlated. The high correlation is explained by the fact that these two spaces are separated by only one affine transformation and without any non-linearities. Note that this test is only a guiding experiment to help find out to what FID is sensitive to and it is not guaranteed to hold for different GAN architectures or training setups.

## C    WHAT DOES FID LOOK AT IN AN IMAGE?

**Validation of FID sensitivity heatmaps.** To validate how reliably our sensitivity heatmaps highlight the most important regions for FID, we perform an additional experiment where we add Gaussian noise to either important or unimportant areas, while keeping the other clean without noise. To

cancel out effects that may arise from adding different amounts of noise, measured in pixel area, we divide the pixels of the images equally between the important and unimportant regions.

Figure 10a shows FID when we add an increasing amount of Gaussian noise to different regions of the images and Figure 10b demonstrates the appearance of the noisy images. Adding noise everywhere in the image is an upper bound how greatly FID can increase in this test setup. Adding noise to the important regions leads to larger increase in FID, compared to adding noise to the unimportant regions.

**Additional FID sensitivity heatmaps.** Figure 11 presents more FID sensitivity heatmaps for individual StyleGAN2 generated images using FFHQ and LSUN CAT and their corresponding ImageNet Top-1 classifications in the top left corner. For both datasets the regions for which FID is the most sensitive to are highly localized and correlate strongly with the Top-1 class.

Figure 12 shows additional mean images and heatmaps for StyleGAN2 generated images for FFHQ that get classified to a certain class. On average FID is the most sensitive to the pixel locations where the Top-1 class is intuitively located and relatively insensitive to the human faces.

**Comparison to Grad-CAM.** Figures 13 and 14 compare our FID sensitivity heatmaps to standard Grad-CAM heatmaps (Selvaraju et al., 2017) for FFHQ and LSUN CAT, respectively. Grad-CAM heatmaps, computed using classification probabilities, highlight similar regions in the images as our FID sensitivity heatmaps, showing that the important regions for ImageNet classification overlap heavily with regions that are important for FID. Additionally, Figure 15 shows mean images and FID heatmaps, as well as mean Top-1 Grad-CAM heatmaps for StyleGAN2 generated FFHQ images.

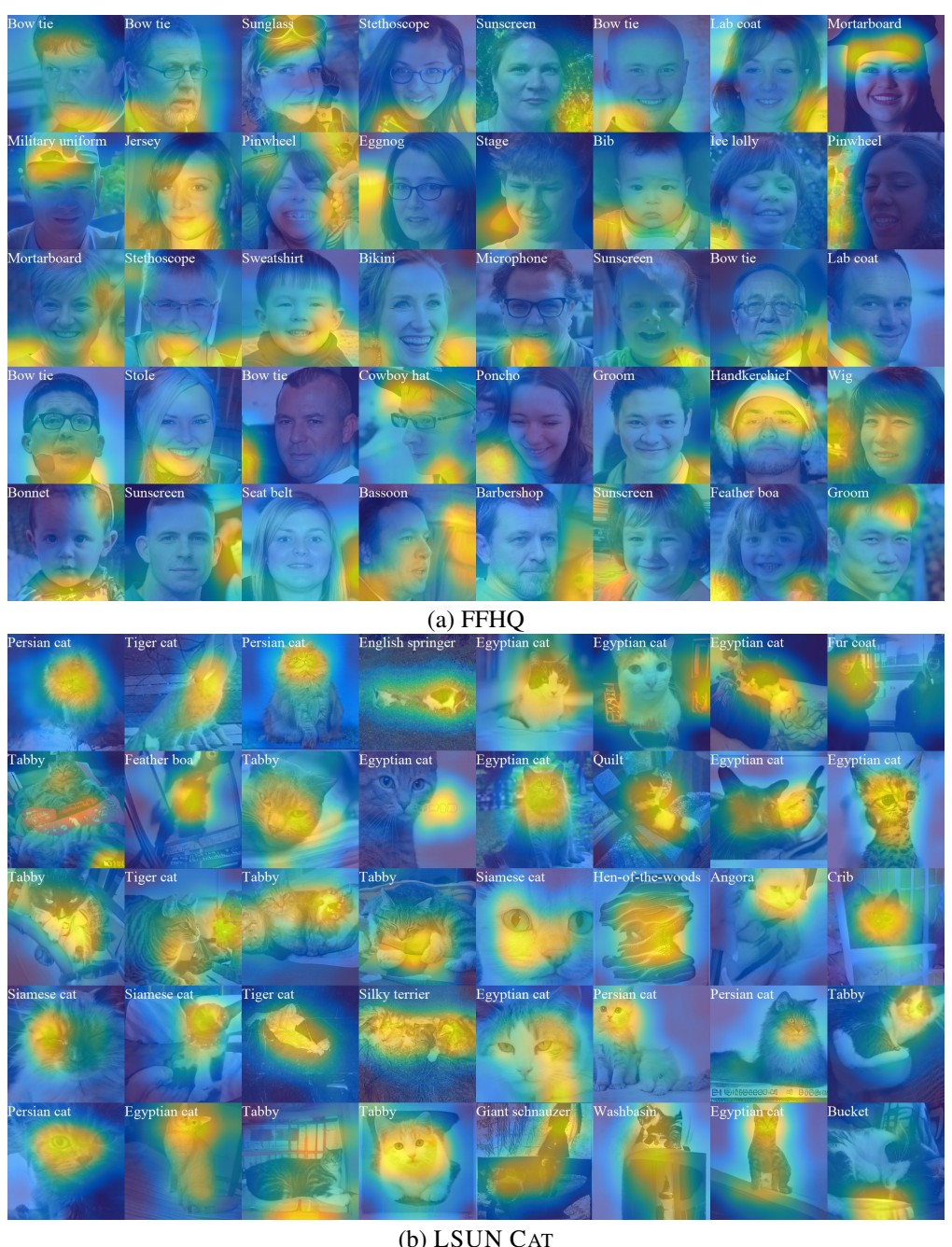

(a) FFHQ

(b) LSUN CAT

Figure 11: Heatmaps of the most important regions for FID for StyleGAN2 images in (a) FFHQ and (b) LSUN CAT, along with their Top-1 classification annotated in the top left corner of each image.

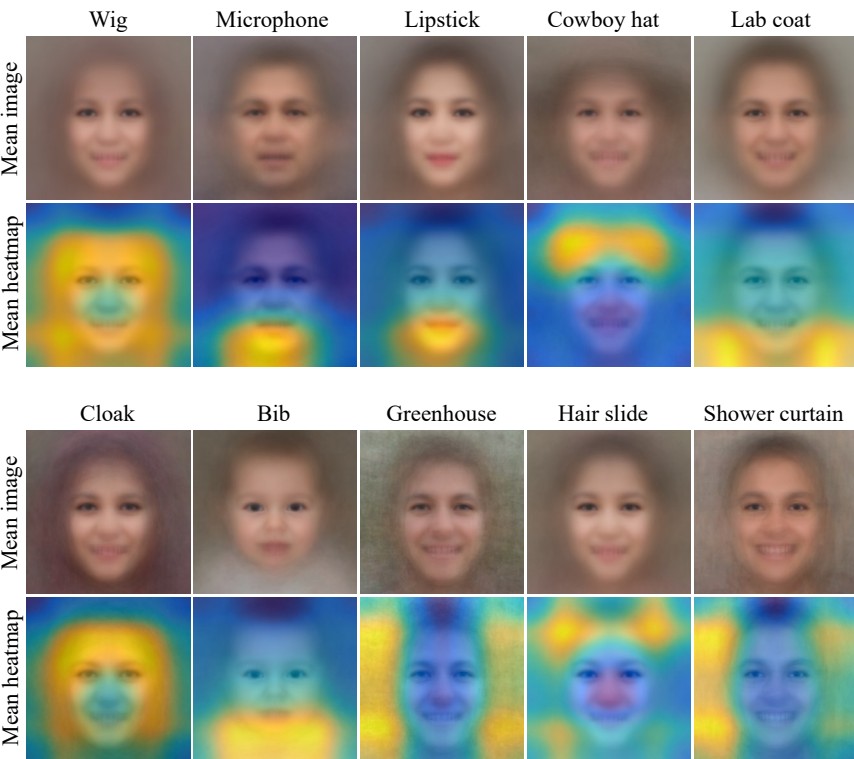

Figure 12: Mean images and heatmaps of regions that are the most important for FID with Style-GAN2 images in FFHQ that get classified to some class, e.g., "lipstick". The heatmaps highlight the regions of Top-1 classes that are typically located outside the face area.

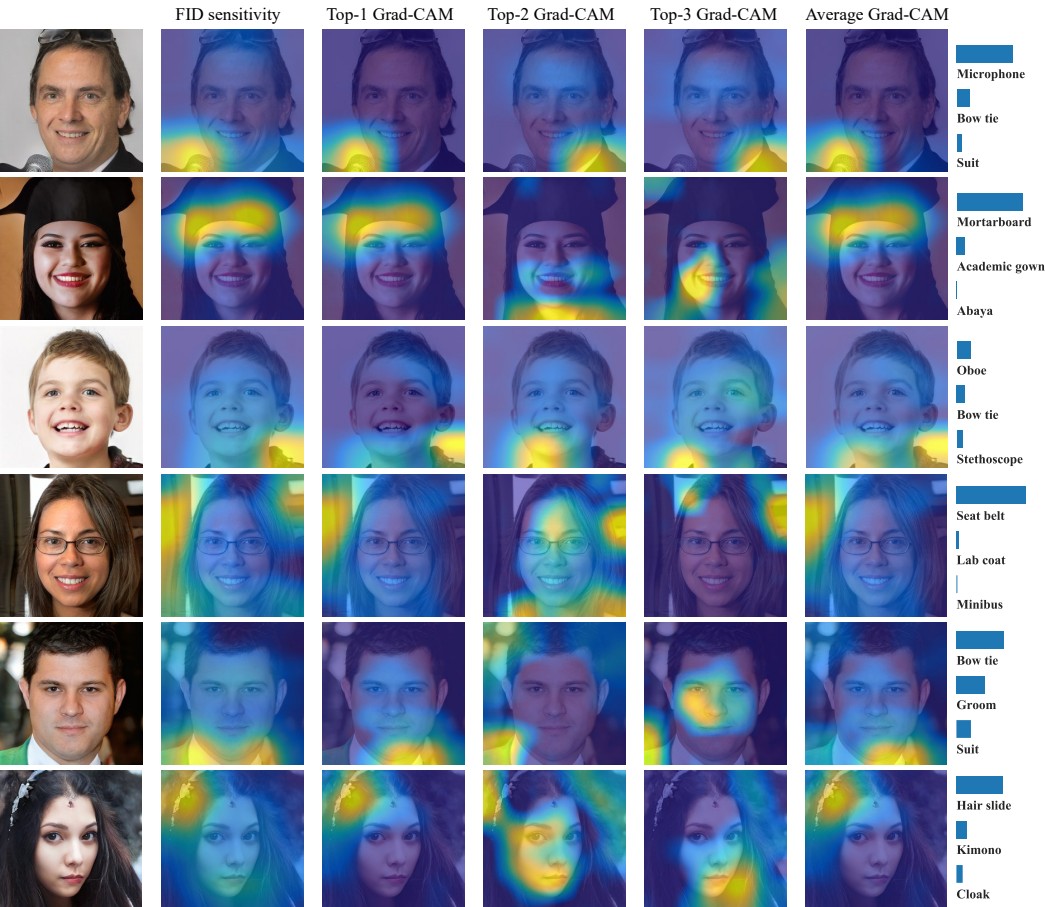

Figure 13: Comparison of our FID sensitivity heatmaps with standard Grad-CAM in FFHQ. The Grad-CAM heatmaps highlight the most important areas for Top-1, Top-2, and Top-3 classification. We also show an average Grad-CAM heatmap weighted according to the classification probabilities.

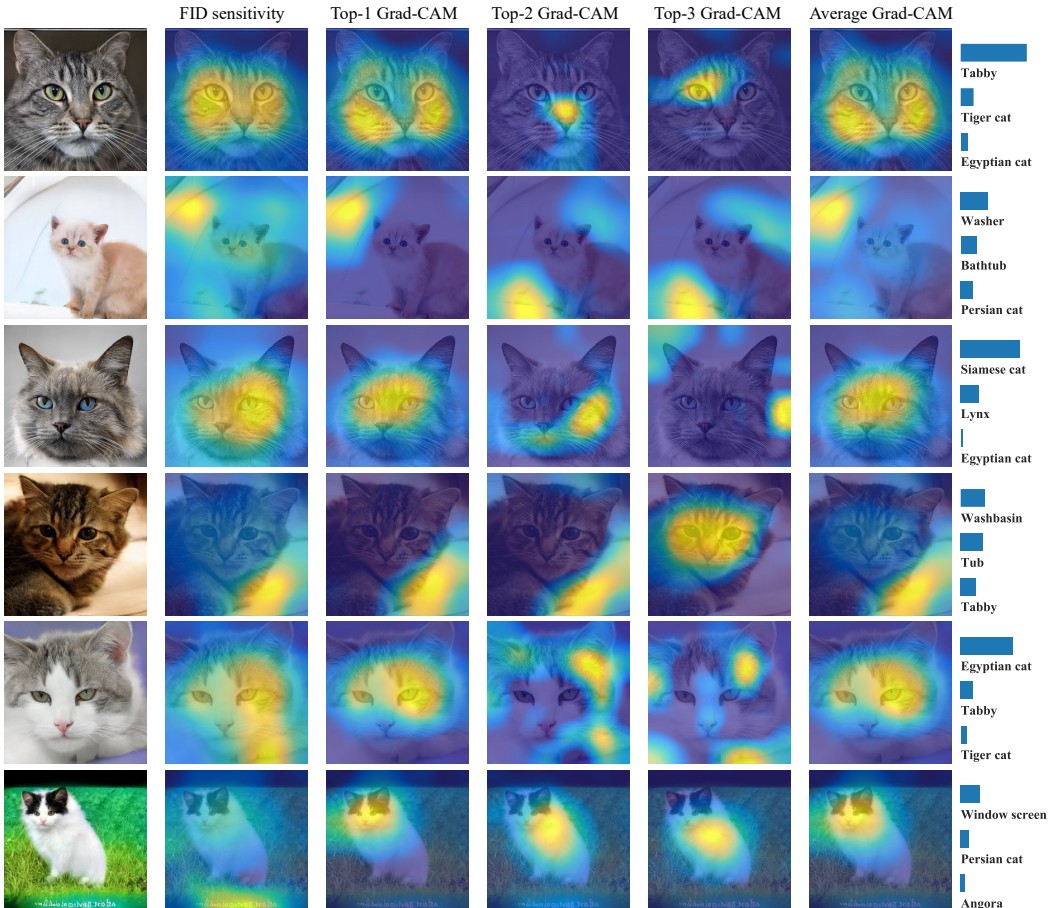

Figure 14: Comparison of our FID sensitivity heatmaps with standard Grad-CAM in LSUN CAT. The Grad-CAM heatmaps highlight the most important areas for Top-1, Top-2, and Top-3 classification. We also show an average Grad-CAM heatmap weighted according to the classification probabilities.

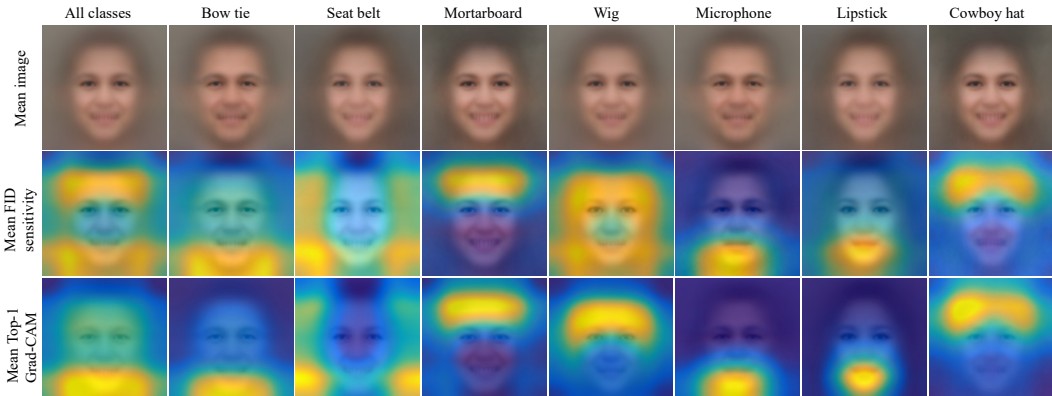

Figure 15: Top: Average of StyleGAN2-generated FFHQ images whose Top-1 classification matches the given class, e.g., "bow tie". Middle: Average heatmaps of regions that are the most important for FID. Bottom: Corresponding average Grad-CAM heatmaps computed for the Top-1 class.

# D   PROBING THE PERCEPTUAL NULL SPACE IN FID

**Pseudocode and implementation details.** Algorithm 1 shows the pseudocode for our resampling method. Function OPTIMIZE-RESAMPLING-WEIGHTS optimizes the per-image sampling weights such that FID between real and weighted generated features is minimized. The inputs to the function are sets of features for real and generated images $\boldsymbol{F}_r$ and $\boldsymbol{F}_g$, respectively, learning rate $\alpha$ and maximum number of iterations $T$. Note that the features do not have to be the typical pre-logits features where standard FID is calculated; they can be, e.g., logits or binarized class probabilities. First, we calculate the statistics of real features (lines 3-4) and then initialize the per-image log-parameterized weights $w_i$ to zeros (line 7). Then, for $T$ iterations we calculate the weighted mean and covariance of generated features (lines 11-12) and update the weights via gradient descent to minimize FID (line 15). After optimization the log-parameterized weights can be transformed into sampling probabilities with $p_i = \frac{e^{w_i}}{\sum_j e^{w_j}}$, where $p_i$ is the probability of sampling $i$th feature. We sample with replacement according to these probabilities to calculate our resampled FIDs (FID$^{PL}$, FID$^L$, FID$^{PL}_{CLIP}$) with 50k real and generated features.

In practice, we use features of 50k real and 250k generated images for all datasets, except for AFHQ-V2 DOG where we use 4678 real and 25k generated images. We use learning rate $\alpha = 10.0$ when optimizing pre-logits features and $\alpha = 5.0$ when optimizing logits or binarized class probabilities. We optimize the weights until convergence, which typically requires $\sim$100k iterations. We select the weights that lead to the smallest FID with 50k real and 50k generated features that are sampled according to the optimized weights. In the optimization, we do not apply exponential moving average to the weights or learning rate decay. We use 32GB NVIDIA Tesla V100 GPU to run our resampling experiments. One weight optimization run with 50k real and 250k generated features takes approximately 48h where most the execution time goes into calculating the matrix square root in FID with eigenvalue decomposition. Code is available at `https://github.com/kynkaat/role-of-imagenet-classes-in-fid`.

**Image grids for Top-1 matching and pre-logits resampling.** Figure 16 shows uncurated image grids when we sample StyleGAN2 generated images randomly, after Top-1 histogram matching, and after matching all fringe features. Even though FID drops very significantly, the visual appearance of the generated images remains largely unchanged. FID$_{CLIP}$ also fails to confirm the improvement indicated by FID.

In Figure 17, we show a larger set of images that obtain a small or large weight after optimizing FID in the pre-logits feature space. A low FID after resampling cannot be attributed to simply removing images with clear visual artifacts.

**Effect of pre-logits resampling on KID.** Table 3 shows that resampling in the pre-logits feature space also strongly decreases Kernel Inception Distance (KID) (Binkowski et al., 2018), and a Kernel Inception Distance that is calculated using the radial basis function (RBF) kernel (RBF-KID). While the standard KID compares the first three moments (Binkowski et al., 2018), RBF-KID considers *all* moments, because the RBF kernel is a characteristic kernel (Gretton et al., 2012; Fukumizu et al., 2007). The metrics are computed in the same feature space as FID and therefore we hypothesize that they share approximately the same perceptual null space. To calculate RBF-KID, we used RBF scatter parameter $\gamma = \frac{1}{d}$, where $d = 2048$ is the dimensionality of Inception-V3 pre-logits. We experimented with different scatter parameter values ($\gamma \in \left\{ \frac{1}{8d}, \frac{1}{4d}, \frac{1}{2d}, \frac{1}{d}, \frac{2}{d}, \frac{4}{d}, \frac{8}{d} \right\}$) and observed that they all lead to similar qualitative behavior.

**Top-$N$ histogram matching**. We show further results from approximate Top-$N$ histogram matching in LSUN CAT/CAR/PLACES and AFHQ-V2 DOG in Figure 18. FID can be consistently improved by aligning the Top-$N$ histograms of real and generated images. Furthermore, the largest decrease in FID can be obtained by including information of the most probable classes.

# E   PRACTICAL EXAMPLE: IMAGENET PRE-TRAINED GANS

Figure 19 shows larger image grids for StyleGAN2 and Projected FastGAN in FFHQ.

---
**Algorithm 1** Resampling algorithm pseudocode.

---
1: **function** OPTIMIZE-RESAMPLING-WEIGHTS($\boldsymbol{F}_\mathrm{r}, \boldsymbol{F}_\mathrm{g}, \alpha, T$)
2:     Calculate feature statistics of reals.
3:     $\boldsymbol{\mu}_\mathrm{r} \leftarrow \frac{1}{|\boldsymbol{F}_\mathrm{r}|} \sum_i \boldsymbol{f}_\mathrm{r}^i$
4:     $\boldsymbol{\Sigma}_\mathrm{r} \leftarrow \frac{1}{|\boldsymbol{F}_\mathrm{r}|-1} \sum_i \left(\boldsymbol{f}_\mathrm{r}^i - \boldsymbol{\mu}_\mathrm{r}\right)^T \left(\boldsymbol{f}_\mathrm{r}^i - \boldsymbol{\mu}_\mathrm{r}\right)$
5:
6:     Initialize log-parameterized per-image weights $\boldsymbol{w}$ to zeros.
7:     $w_i = 0, \ \forall i$
8:
9:     **for** $T$ iterations **do**
10:         Compute weighted mean and covariance of generated features.
11:         $\boldsymbol{\mu}_\mathrm{g}(\boldsymbol{w}) \leftarrow \frac{\sum_i e^{w_i} \boldsymbol{f}_\mathrm{g}^i}{\sum_i e^{w_i}}$
12:         $\boldsymbol{\Sigma}_\mathrm{g}(\boldsymbol{w}) \leftarrow \frac{1}{\sum_i e^{w_i}} \sum_i e^{w_i} \left(\boldsymbol{f}_\mathrm{g}^i - \boldsymbol{\mu}_\mathrm{g}(\boldsymbol{w})\right)^T \left(\boldsymbol{f}_\mathrm{g}^i - \boldsymbol{\mu}_\mathrm{g}(\boldsymbol{w})\right)$
13:
14:         Update the weights.
15:         $\boldsymbol{w} \leftarrow \boldsymbol{w} - \alpha \nabla_{\boldsymbol{w}} \mathrm{FID}\left(\boldsymbol{\mu}_\mathrm{r}, \boldsymbol{\Sigma}_\mathrm{r}, \boldsymbol{\mu}_\mathrm{g}(\boldsymbol{w}), \boldsymbol{\Sigma}_\mathrm{g}(\boldsymbol{w})\right)$
16:
17:     **return** $\boldsymbol{w}$

---

Table 3: Optimizing FID also decreases KID and RBF-KID significantly. We compare the KIDs of randomly sampled images (KID, RBF-KID) against KIDs computed by resampling according to the weights obtained from optimizing FID in the pre-logits features (KID$^{\mathrm{PL}}$, RBF-KID$^{\mathrm{PL}}$). The numbers represent averages over ten evaluations.

| Dataset | FID | FID$^{\mathrm{PL}}$ | KID$\times 10^3$ | KID$^{\mathrm{PL}}$ $\times 10^3$ | RBF-KID$\times 10^3$ | RBF-KID$^{\mathrm{PL}}$ $\times 10^3$ | |
|---------|-----|---------|---------|---------|---------|---------|---|
| FFHQ | 5.30 | 1.78 $(-66.4\%)$ | 1.52 | 0.19 $(-87.5\%)$ | 0.67 | 0.08 | $(-88.1\%)$ |
| LSUN CAT | 8.25 | 3.05 $(-63.0\%)$ | 3.00 | 0.37 $(-87.7\%)$ | 1.32 | 0.16 | $(-87.9\%)$ |
| LSUN CAR | 5.65 | 2.11 $(-62.7\%)$ | 2.49 | 0.32 $(-87.1\%)$ | 1.19 | 0.16 | $(-86.6\%)$ |
| LSUN PLACES | 12.96 | 3.59 $(-72.3\%)$ | 7.43 | 0.31 $(-95.8\%)$ | 3.04 | 0.14 | $(-95.4\%)$ |
| AFHQ-v2 DOG | 10.25 | 5.92 $(-42.2\%)$ | 2.05 | 0.12 $(-94.1\%)$ | 1.04 | 0.06 | $(-94.2\%)$ |

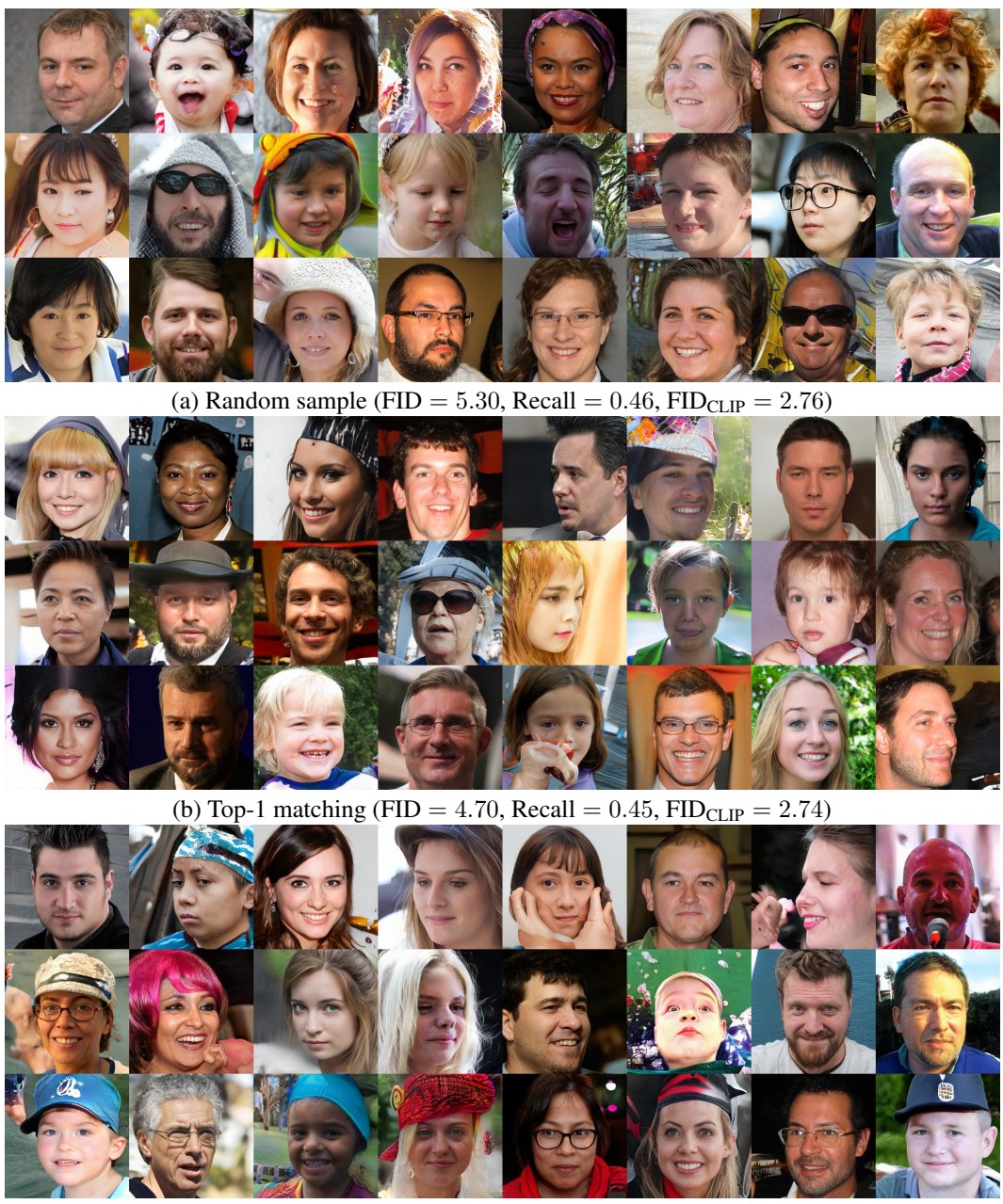

(a) Random sample (FID = 5.30, Recall = 0.46, $FID_{CLIP}$ = 2.76)

(b) Top-1 matching (FID = 4.70, Recall = 0.45, $FID_{CLIP}$ = 2.74)

(c) Pre-logits resampling (FID = 1.78, Recall = 0.40, $FID_{CLIP}$ = 2.64)

Figure 16: FID can be drastically reduced by using our resampling approach without improving the visual fidelity of the generated images in any obvious way. (a) Randomly sampled StyleGAN2 images (b) Randomly sampled StyleGAN2 images after Top-1 histogram matching (c) StyleGAN2 images sampled according to the weights obtained by matching all fringe features.

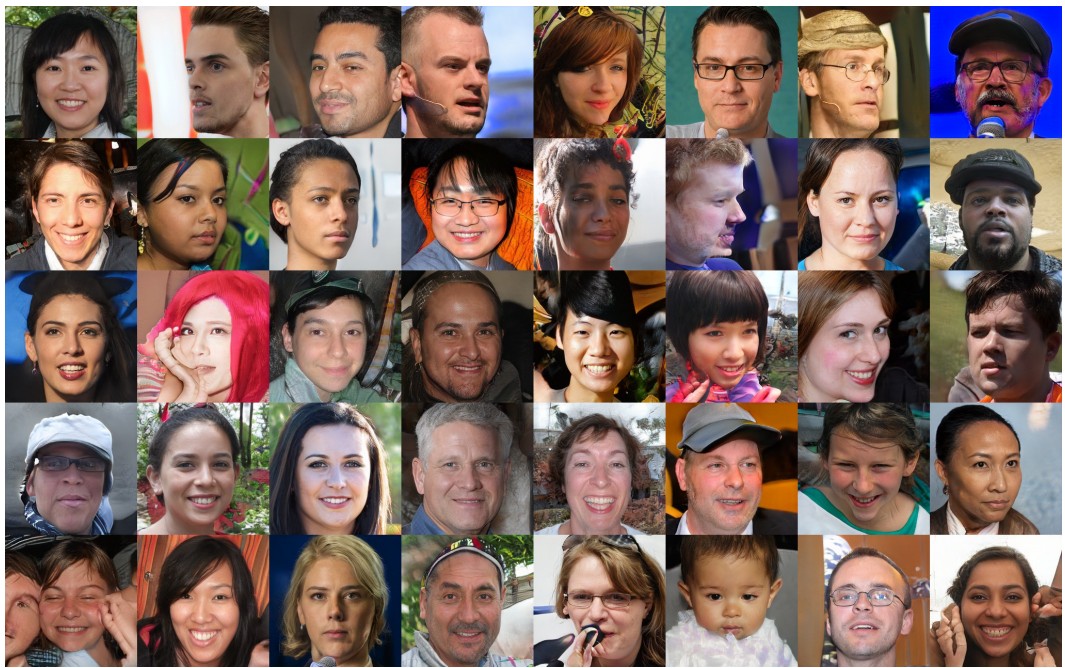

(a) Images with small weights

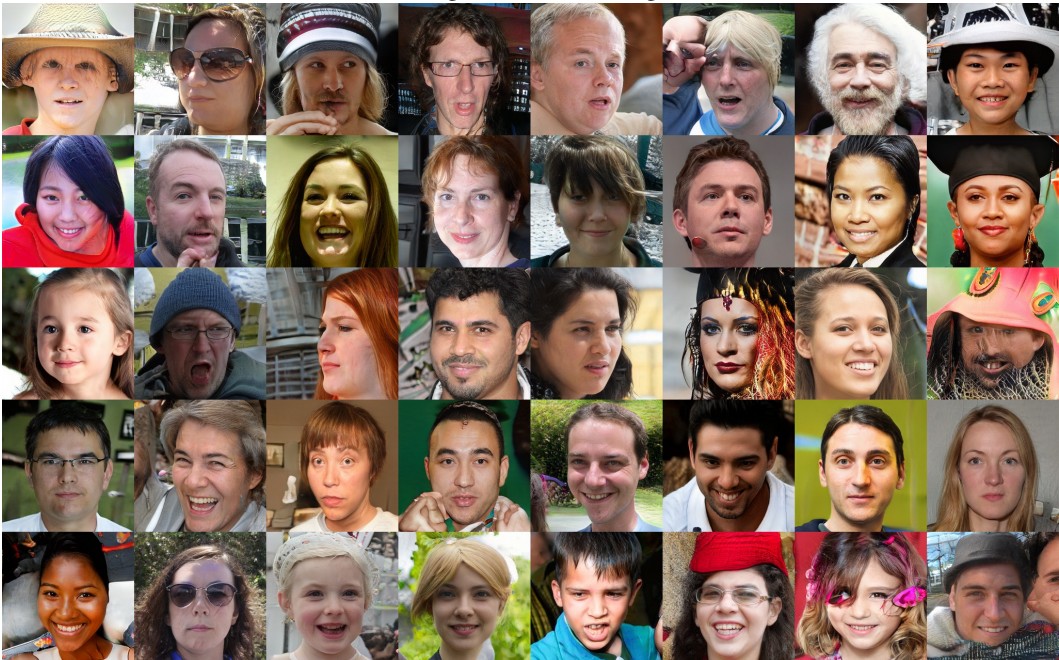

(b) Images with large weights

Figure 17: Random StyleGAN2 images sampled among (a) the smallest 10% of weights and (b) the largest 10% of weights. Both sets contain realistic looking images and images with visual artifacts.

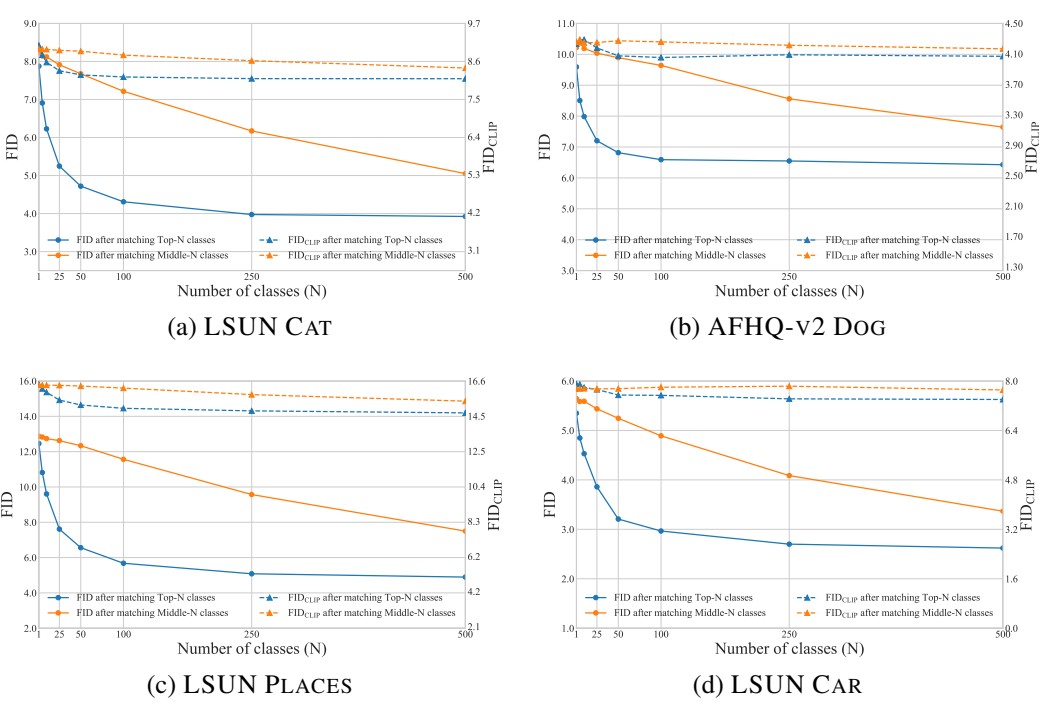

(a) LSUN Cat

(b) AFHQ-v2 Dog

(c) LSUN Places

(d) LSUN Car

Figure 18: Additional results from aligning Top-N class histograms. For all datasets adding information from the Top-N ImageNet classes consistently leads to the largest decrease in FID while FID$_{\text{CLIP}}$ remains almost unchanged.

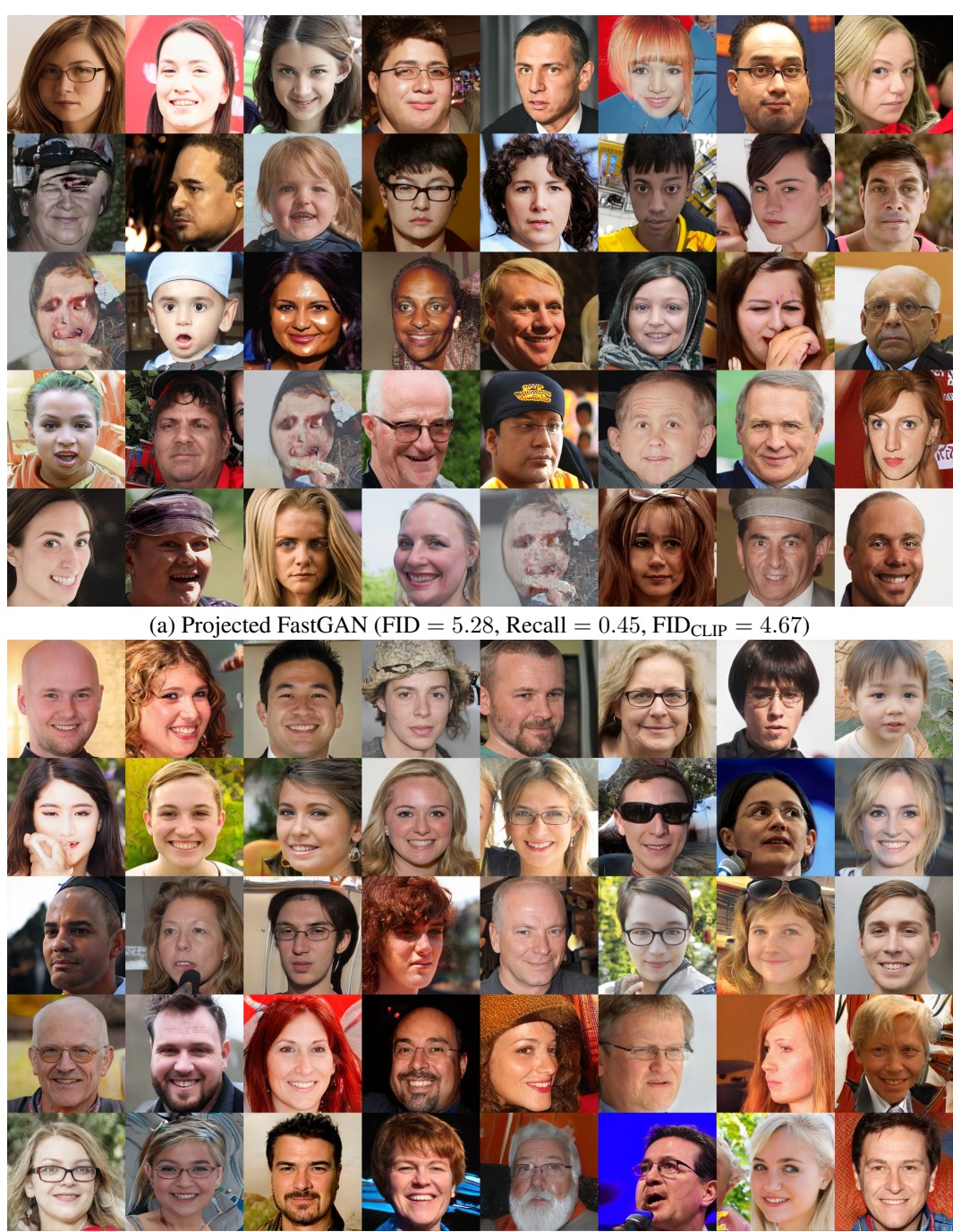

(a) Projected FastGAN (FID = 5.28, Recall = 0.45, FID$_{\text{CLIP}}$ = 4.67)

(b) StyleGAN2 (FID = 5.30, Recall = 0.46, FID$_{\text{CLIP}}$ = 2.76)

Figure 19: Uncurated samples of (a) Projected FastGAN and (b) StyleGAN2 generated images. While Projected FastGAN achieves a better FID, the samples contain more distortions and artifacts.

