# OpenReview forum: "The Role of ImageNet Classes in Fréchet Inception Distance"
_ICLR.cc/2023/Conference — ICLR 2023 notable top 25%_

### Official Review · Reviewer_iTAh · 2022-10-15

**Confidence:** 3
**Clarity, Quality, Novelty And Reproducibility:** This is a high-quality paper, novel, …
**Correctness:** 4
**Technical Novelty And Significance:** 4
**Empirical Novelty And Significance:** 4
**Recommendation:** 8

**Strength And Weaknesses:**

Strength:
1. This paper provides an insightful study of the popular FID score used in GAN, and shows that using ImageNet pre-trained features can be biased. This is new knowledge.

2. Paper is clearly written. The reader can quickly understand the major lesson.


**Summary Of The Paper:**

This paper conducts a deep, detailed study using the FID score for evaluating the quality of generative models. This paper reveals that imagenet pretrained model feature-based metric can be biased, leading to low generation quality with high FID score if the generator aligns the histograms of Top-N classifications be- tween sets of generated and real images. Numerical results and visualizations support the claim.

**Summary Of The Review:**

Good paper, I recommend for acceptance.

---

> ### Author Response · Authors · 2022-11-17
> **Authors' response to reviewer iTAh**
>
> Thank you for the review.

---

### Official Review · Reviewer_9T7x · 2022-10-17

**Confidence:** 4
**Correctness:** 4
**Technical Novelty And Significance:** 2
**Empirical Novelty And Significance:** 2
**Recommendation:** 6

**Clarity, Quality, Novelty And Reproducibility:**

Clarity: The manuscript is clearly written, pleasant to read, and easy to follow.

Quality: Hypothesis are clearly stated, the experimental design is clearly described, and results are discussed in the light of the posed hypothesis, which I think makes it for a high quality empirical work.

Novelty: The results are novel to my knowledge although to me it seems the conclusions pretty much align with the common sense in the community in that FID can be useful but not fully relied upon. Authors clearly showed that to be indeed the case.

Reproducibility: The experimental design is clearly discussed and it covers openly available data and models.

**Strength And Weaknesses:**

Pros:

+Authors focused on a highly relevant problem: evaluating evaluation metrics themselves.

+A somewhat large-scale empirical evaluation is carried out to reveal issues in the so commonly used Fréchet Inception Distance: It is noisy/high variance when used in data that somehow differs from ImageNet-1k.

+More interestingly, authors show evidence for what they called the perceptual null space. I.e, FID is exploitable and one can modify it to a great extent without imposing perceptual improvement in generated samples.

Cons:

-My main concern with the paper is the size of its scope, which is a subjective concern rather than a technical one admittedly. While the work defines clear hypothesis involving the properties of FID and provides clear evidence supporting those, it lacks in providing solutions or the means to address the observed issues. Authors did observe that using self-supervised backbones, for instance, would alleviate the issue, but that isn't given enough focus and it is not clear whether we should move towards this kind of solution.

-Another important concern which is inline with the above is the scope of the empirical assessment. While it clearly supported the main claims and gives a clear demonstration of the issues of FID, it's limited to that. Some lacking components I would mention would be:

1-Studying the effect of using backbones trained with approaches other than supervision on ImageNet-1K. I would expect that simply moving from training the encoder on ImageNet-1K to a dataset with a larger label set, e.g., ImageNet-21K, would alleviate observed issues quite significantly, and render "attacks" much less efficient to execute. A similar effect could be obtained with training paradigms other than supervision, as indicated by some of the reported results. So, it would be very informative to repeat some of the experiments with a self-supervised backbone trained on a highly diverse dataset (i.e., with a large label set).

2-Verifying whether features relevant for FID indeed correspond those relevant for supervision. I would suggest the authors somehow compare the FID heatmaps they proposed with standard GradCAM. Do those match?

3-Studying whether the same behaviour in FID is observed with KID under varying kernels. FID only accounts for the first two moments while KID, under the right kernel, should account for high order moments as well. Would accounting for those extra moments help overcome the issue with FID or would they make it for an even more vulnerable metric? The results in Appendix D indicate the latter, but only an analysis under varying kernels would be conclusive.




**Summary Of The Paper:**

In this paper, authors carry out an extensive empirical assessment of a popular evaluation metric for generative models of natural images: the Fréchet Inception Distance. In particular, experiments are designed to indicate properties of the distance when used on data that diverges somehow from ImageNet-1k, used to pre-train the Inception architecture used as an embedding mapping to compress images prior to computing the metric. Results indicate that, under such distribution shifts, the Fréchet Inception Distance can be misleading as it is affected by features that are not as relevant in the downstream data as they were when supervisedly training the Inception encoder. Importantly, results revealed that generated samples tend to reduce the metric when they match the top classes appearing in the subset of ImageNet used to compute the metric, which can be exploited using simple resampling procedures, as demonstrated in the paper.

**Summary Of The Review:**

In summary, I would describe the paper as a solid empirical assessment of an important problem, but lacking the proposal of solutions, which, in my opinion, limit too much the scope of the paper. I would be more than happy to bump up my scores if the authors are able to introduce results that are more indicative of improving directions and leave less for future work.

---

> ### Author Response · Authors · 2022-11-17
> **Authors' response to reviewer 9T7x**
>
> Thank you for the review. We will address the explicit questions:
>
> Q: “Verifying whether features relevant for FID indeed correspond those relevant for supervision. I would suggest the authors somehow compare the FID heatmaps they proposed with standard GradCAM. Do those match?”
>
> Thank you for the great suggestion. We extended the supplement with comparisons between the FID heatmaps and the standard Grad-CAM heatmaps (Figures 13, 14, and 15). They show that Grad-CAM heatmaps, which are slightly more localized, highlight similar areas as FID heatmaps. This further emphasizes that the features relevant for FID correspond to features that are important for ImageNet classification.
>
> Q: “Studying whether the same behaviour in FID is observed with KID under varying kernels. FID only accounts for the first two moments while KID, under the right kernel, should account for high order moments as well. Would accounting for those extra moments help overcome the issue with FID or would they make it for an even more vulnerable metric? The results in Appendix D indicate the latter, but only an analysis under varying kernels would be conclusive.”
>
> We extended Table 3 in our supplement to include numbers from KID that uses the RBF kernel, which accounts for higher order moments as well. RBF-KID decreases just as strongly as the standard KID when we optimize FID. We also computed a sweep over different values for the RBF kernel spread parameter and confirmed that qualitatively similar behavior results for all of them.
>
> Thank you for suggesting these interesting experiments. The newly added passages are denoted using a green color in the PDF.
>
> Q: “Studying the effect of using backbones trained with approaches other than supervision on ImageNet-1K. I would expect that simply moving from training the encoder on ImageNet-1K to a dataset with a larger label set, e.g., ImageNet-21K, would alleviate observed issues quite significantly, and render "attacks" much less efficient to execute. A similar effect could be obtained with training paradigms other than supervision, as indicated by some of the reported results. So, it would be very informative to repeat some of the experiments with a self-supervised backbone trained on a highly diverse dataset (i.e., with a large label set).”
>
> While we agree that some alternative feature space and/or training protocol is likely to improve the situation markedly, we also feel that such an investigation should be carried out in detail that is not possible within the ICLR page limit without losing a significant fraction of our results; we consciously chose to focus on the common case of Inception-V3/ImageNet-1K and present a broad set of experiments that highlight its unintuitive properties. As ImageNet-1k classifiers are so widely used in pretrained models (as well as in quality metrics), we believe that deeper understanding of their behavior and caveats is of considerable practical significance.

---

> > ### Comment · Reviewer_9T7x · 2022-11-23
> > **Response to authors.**
> >
> > Thank you for the extra experiments. I raised my score to 6 since I still think that the scope is a bit limited if not enough focus is given to proposing an alternative to FID that solves the observed issues.
> >
> > As a final comment, based on the extra results with KID, it seems that adding higher order moments doesn't help much. Perhaps dropping moments could help then, so a variation of FID where only the first-order moments are considered could be less prone to attacks than the standard version.

---

### Official Review · Reviewer_ncn9 · 2022-10-18

**Confidence:** 4
**Correctness:** 4
**Technical Novelty And Significance:** 2
**Empirical Novelty And Significance:** 3
**Recommendation:** 5

**Clarity, Quality, Novelty And Reproducibility:**

Generally, the paper has clear claims and a nice experimental setup supporting them. I have a concern regarding the novelty of the main contribution of the work conveyed in the first weakness. I hope the authors provide a clear response for that regard.

**Strength And Weaknesses:**

Strengths:

1- The problem tackled in this paper is important: Metrics used to assess the quality of generative models need to be reliable.
The results of this paper suggests that one could improve such metrics without actual improvements in the perceptual quality of the generative model.

2- Constructing an adversarial attack to FID through matching the Top-1/Top-N histograms of the output space between the generated distribution of images and ImageNet is both novel and insightful.

3- The paper well motivates the experimental setup through showing the sensitivity of FID to each pixel in a given image. Experiments in Figure 3 show that FID could pay more attention to non-semantic parts of the generated images.


Weaknesses:

1- While this paper proposes very interesting analysis of FID, most of the conclusions of this paper are not surprising. For example, [A] discussed the potential of constructing adversarial attacks to the inception score. Moreover, [B] showed that one could generate images of noise that has good FID scores. Further, they show that  FID favors distribution of images with more artifacts by adjusting the truncation level in StyleGAN2.

It would be ideal to distinguish the new insights that this paper brings over the previous analysis of GAN metrics.

2- Despite that this paper is mainly analyzing FID in the lens of ImageNet pre training, I believe that proposing a step towards fixing the spotted issue in FID is necessary.
The results of FID_{CLIP} presented towards the end of the paper are promising.
Can you conduct some of the following experiments to validate the usefulness of such metric:

2.a- Compute FID_{CLIP} for several GANs in the literature such as StyleGANV1, StyleGANV2, StyleGANV2, Big GAN. Does FID_{CLIP} order these GANs in the same way their perceptual quality order?

2.b- Does FID_{CLIP} separate , for instance, a distribution of images from a distorted one (e.g. blurring and Gaussian noise).

2c- Is FID_{CLIP} better due to a better architecture, Better pre training, or both? For instance, if we assume we have a subset of the training set of CLIP available, can we fool FID_{CLIP} with a similar approach to the one presented in this paper?

Generally, I appreciate the efforts put in this work. I see several merits despite the aforementioned weaknesses which I hope to be addressed during the discussion.

[A]: “Improved Techniques for Training GANs”, 2016

[B]: “On the robustness of Quality Measures for GANs”, 2022

**Summary Of The Paper:**

This paper presents a careful evaluation of on of the widely used metrics in the generative models literature; Fréchet Inception Distance (FID). The experiments demonstrates and equivalence between the feature of Inception V3 module space in which FID is computed and the logits. That is, FID is mainly sensitive towards features that correlates with categories in the ImageNet dataset. Further, several experiments were carried out to match the Top-1 and Top-N output distribution of generated images with the ImageNet one. The results show that one can significantly improve FID without actual improvement in the image quality. At last, experiments in Figure 7 show promising results of computing FID in the CLIP embedding space that might be more generic than ImageNet pre training,.

**Summary Of The Review:**

This paper has several merits including the motivation and experimental setup. However, I have two main concerns regarding the novelty and the a fix to the spotted issue in FID.

---

> ### Author Response · Authors · 2022-11-17
> **Authors' response to reviewer ncn9**
>
> Thank you for the review. We will address the explicit questions and requests:
>
> Q: “It would be ideal to distinguish the new insights that this paper brings over the previous analysis of GAN metrics. [...] Moreover, [B] showed that one could generate images of noise that has good FID scores. Further, they show that FID favors distribution of images with more artifacts by adjusting the truncation level in StyleGAN2.”
>
> We acknowledge that scattered evidence exists in the literature that FID is biased towards ImageNet classes and therefore can yield unreliable results. However, to the best of our knowledge, the topic has not received a systematic investigation before. We fill this gap by showing _what_ FID measures from the generated images, _why_ it can be unreliable in some cases, and also demonstrate that this has practical consequences (e.g., pre-training in GANs).
>
> Thank you for pointing out the missing citation [B]. The orthogonal work in [B] shows that FID can be significantly manipulated by modifying the _pixel contents of the images_, i.e., using a standard adversarial attack. However, generative models do not use FID as a loss function, and thus it is not immediately obvious how to apply this result in practice, or how it relates to the ImageNet classes. Our histogram matching is a much weaker form of attack as it does not change the pixel values at all, and it is designed to elucidate the practical behavior seen in generative models.
>
> We hypothesize that the observation in [B] — that FID can favor a distribution of images with more artifacts — follows from lower truncation leading to improved diversity, thus better matching the training distribution. After all, FID quantifies the alignment of distributions, not the quality of individual images alone.
>
> Q: “Despite that this paper is mainly analyzing FID in the lens of ImageNet pre training, I believe that proposing a step towards fixing the spotted issue in FID is necessary.”
>
> We wholeheartedly agree that research is needed on new metrics that circumvent the observed shortcomings in FID. However, we feel this is not feasible within the page limit of an ICLR paper without sacrificing much of the present analysis, and believe that the audience of generative model researchers like ourselves will value the breadth of observations presented herein. Alternative feature spaces and/or training regimes would warrant a separate, equally thorough investigation to validate their overall robustness and probe the qualitative and quantitative nature of their respective perceptual null spaces. In particular, question 2 and its sub-items would be exactly the kinds of experiments that would be valuable to conduct in such future work.
>
> As ImageNet and FID (as currently defined) are used so widely, we believe that deeper understanding of their behavior and caveats is of considerable practical significance.
>
> Q: “Is FID_{CLIP} better due to a better architecture, Better pre training, or both? For instance, if we assume we have a subset of the training set of CLIP available, can we fool FID_{CLIP} with a similar approach to the one presented in this paper?”
>
> Thank you for the interesting question. In the paper, we use FID_{CLIP} as a control to assess whether a superfluous drop in FID is picked up by another feature space. The negative result does not imply that FID_{CLIP} would be “better”. Indeed, we have tried optimizing FID_{CLIP} by resampling, and observed that it can also be improved in an equally dramatic magnitude as FID, without clear differences in the original vs. resampled image sets. This leads us to hypothesize that it, too, has a significant perceptual null space – just not the same as that of FID. As stated above, we believe that the properties of this and other potential feature spaces warrant an equally comprehensive study in future work.

---

### Official Review · Reviewer_Xd21 · 2022-10-21

**Confidence:** 3
**Correctness:** 4
**Technical Novelty And Significance:** 4
**Empirical Novelty And Significance:** 4
**Recommendation:** 8

**Clarity, Quality, Novelty And Reproducibility:**

This paper is very valuable and overall clearly written. The finding and discussion about the important metric FID are very meaningful. The visualizations are convincing and insightful.

**Strength And Weaknesses:**

This paper is well organized, with reasonable structure and clear logic. My concerns are as follows.

1. In page 6 (bottom), the paper mention the first step adopt ''5×oversampling''. Why you choose ''5×oversampling''? How does this parameter affect the match process?
2. In Table 1, FID_{SwAV} and FID_{CLIP} have ∼ 1% decrease after matching. I'm very curious about the reasons for this decline.



**Summary Of The Paper:**

This work uses visualization techniques to investigate the area of concern of FID in the generated images. Through the study and analysis, the authors have an interesting finding that the feature space computed by FID usually is very close to ImageNet classification. The paper also gives apractical example of an accidental distortion and provides suggestions for viable alternative feature spaces.

**Summary Of The Review:**

This paper explores possible reasons why FID sometimes deviates from human judgment and provides guidance for the subsequent use of FID.

---

> ### Author Response · Authors · 2022-11-17
> **Authors' response to reviewer Xd21**
>
> Thank you for the review. We will address the explicit questions:
>
> “1. In page 6 (bottom), the paper mention the first step adopt “5x oversampling”. Why you choose “5x oversampling”? How does this parameter affect the match process?”
>
> We use oversampling to get a rich set of candidate images, including ones that get classified to rare ImageNet classes. If we reduce the oversampling factor from 5x, the FID drops slightly less (for example in FFHQ, 5x -> FID 1.8, 3x -> FID 2.1), while increasing the factor leads to an even larger drop in FID. On the other hand, larger oversampling factors consume more GPU memory, and we hit a limit with our current implementation around 10x. Thus, we chose 5x oversampling as a tradeoff that clearly shows the existence of perceptual null space while still being easy to compute.
>
> “2. In Table 1, FID_{SwAV} and FID_{CLIP} have ∼ 1% decrease after matching. I'm very curious about the reasons for this decline.”
>
> We find this question interesting and hypothesize that the observed decline in FID_{SwAV} and FID_{CLIP} is because the features that SwAV and CLIP are sensitive to have some amount of overlap with Inception-V3 features: SwAV was trained with ImageNet data, and CLIP can also cover ImageNet categories; zero-shot ImageNet classification using CLIP works reasonably well, after all.

---

### Decision · Program_Chairs · 2023-01-20

**Decision:**

Accept: notable-top-25%

**Justification For Why Not Higher Score:**

While this paper was a strong technical paper, what would push it over the edge for me is if there were clear fixes to the metric.

**Justification For Why Not Lower Score:**

Given how important FID is to the generative modeling community, and given how extensive the empirical evaluation is, I cannot justify a lower score.

**Metareview: Summary, Strengths And Weaknesses:**

The authors conduct an extensive study of the role of ImageNet classes on the resulting FID score. This work is important as FID is the most metric most widely used to evaluate generative models. Through careful analysis, the authors show that one can decrease FID by as much as 60% without improving sample quality.

The strength in the paper is extensive empirical evaluation. The one minor weakness, as pointed out by some reviewers, is the lack of a "fix" of the metric. Still, given the pervasiveness of FID in generative model evaluation, this work is important to the community.

**Note From Pc:**

if the above contains the word "oral" or "spotlight" please see: "oral" presentation means -> notable-top-5% and "spotlight" means -> notable-top-25%. As stated in our emails, we are disassociating presentation type from AC recommendations